# Detection and In Vivo Validation of *Dichorhavirus* e-Probes in Meta-Transcriptomic Data via Microbe Finder (MiFi^®^) Discovers a Novel Host and a Possible New Strain of Orchid Fleck Virus

**DOI:** 10.3390/v17030441

**Published:** 2025-03-19

**Authors:** Avijit Roy, Jonathan Shao, Andres S. Espindola, Daniel Ramos Lopez, Gabriel Otero-Colina, Yazmín Rivera, Vessela A. Mavrodieva, Mark K. Nakhla, William L. Schneider, Kitty Cardwell

**Affiliations:** 1Molecular Plant Pathology Laboratory, Beltsville Agricultural Research Center (BARC), United States Department of Agriculture (USDA)-Agricultural Research Service (ARS), Beltsville, MD 20705, USA; 2Animal Parasitic Diseases Laboratory, Beltsville Agricultural Research Center (BARC), United States Department of Agriculture (USDA)-Agricultural Research Service (ARS), Beltsville, MD 20705, USA; jonathan.shao@usda.gov; 3Institute for Biosecurity and Microbial Forensics, Oklahoma State University, Stillwater, OK 74078, USA; aramosl@okstate.edu (D.R.L.); kitty.cardwell@okstate.edu (K.C.); 4Department of Entomology and Plant Pathology, Oklahoma State University, Stillwater, OK 74078, USA; 5Colegio de Postgraduados, Campus Montecillo, Texcoco 56264, Mexico; gotero@colpos.mx; 6Plant Protection and Quarantine, Science and Technology, Plant Pathogen Confirmatory Diagnostics Laboratory, Animal Plant Health Inspection Service, United States Department of Agriculture (USDA), Laurel, MD 20708, USA; yazmin.rivera@usda.gov (Y.R.); vessela.a.mavrodieva@usda.gov (V.A.M.); mark.knakhla@yahoo.com (M.K.N.); 7F1/K9 LLC, Palm Coast, FL 32164, USA; wlschneider09@yahoo.com

**Keywords:** *Brevipalpus* transmitted virus (BTV), EDNA, MiFi, e-probe, high-throughput sequencing, orchid fleck virus, virus detection, virus discovery

## Abstract

*Dichorhavirus* is a recently accepted plant virus genus within the family *Rhabdoviridae*. Species assigned to the genus consist of bi-segmented, negative sense, single-stranded RNA viruses and are transmitted by *Brevipalpus* spp. Currently, there are five recognized species and two unclassified members in the genus *Dichorhavirus.* Four out of seven-orchid fleck virus (OFV), citrus leprosis virus N, citrus chlorotic spot virus, and citrus bright spot virus-can infect citrus and produce leprosis disease-like symptoms. The E-probe Diagnostic for Nucleic Acid Analysis (EDNA) was developed to reduce computational effort and then integrated within Microbe-Finder (MiFi^®^) online platform to design and evaluate e-probes in raw High Throughput Sequencing (HTS) data. During this study, *Dichorhavirus* genomes were downloaded from public databases and e-probes were designed using the MiProbe incorporated into the MiFi^®^ platform. Three different sizes of e-probes, 40, 60, and 80 nucleotides, were developed and selected based on whole genome comparisons with near-neighbor genomes. For curation, each e-probe was searched in the NCBI nucleotide sequence database using BLASTn. All the e-probes that had hits with non-target species with ≥90% identities were removed. The sensitivity and specificity of *Dichorhavirus* genus, species, strain, and variant-specific e-probes were validated in vivo using HTS meta-transcriptomic libraries generated from *Dichorhavirus*-suspected citrus, orchid, and ornamentals. Through downstream analysis of HTS data, EDNA not only detected the known hosts of OFV but also discovered an unknown host leopard plant (*Farfugium japonicum*), and the possible existence of a new ornamental strain of OFV in nature.

## 1. Introduction

*Rhabdoviridae* is one of the families in the order *Mononegavirales*. Except for unassigned genera, all other taxonomically accepted rhabdoviruses are allocated into three distinct subfamilies: *Alpharhabdovirinae*, *Betarhabdovirinae*, and *Gammarhabdovirinae*. Out of three, the subfamily *Betarhabdovirinae* includes six virus genera infecting plant hosts and arthropod vectors. Genera assigned to the family *Rhabdoviridae* consist of mono- and bi-segmented, negative-sense, single-stranded RNA viruses [1,2]. Very recently, a data mining expedition discovered the first tri-segmented rhabdovirus genome from multiple hosts tentatively named “Trirhavirus” [3]. The first occurrence of Medicago trirhavirus 1 has been confirmed in commercial alfalfa fields in Washington State, USA [4]. There are two genera (*Dichorhavirus* and *Varicosavirus*) under the family *Rhabdoviridae*, having bi-segmented genomes and infecting plants only. Among these, only *Dichorhavirus* genus is transmitted by false spider mites, belonging to the genus *Brevipalpus* [2,5]. Since the discovery of orchid fleck virus (OFV), the type species of the genus *Dichorhavirus*, four more approved members, [coffee ring spot virus (CoRSV), citrus leprosis virus N (CiLV-N), citrus chlorotic spot virus (CiCSV), and clerodendrum chlorotic spot virus (ClCSV)], and two more unassigned species [citrus bright spot virus (CiBSV) and *Dichorhavirus* sp. ‘monocotyledonae’] were included (https://www.ncbi.nlm.nih.gov/taxonomy/?term=*Dichorhavirus*) (accessed on 29 January 2025). Genome segment 1 (RNA1; ~7.0 kb) of the dichorhaviruses consists of five open reading frames (ORFs) that encode the nucleocapsid protein (N), phosphoprotein (P), movement protein (MP), matrix protein (M), and glycoprotein (G). Segment 2 (RNA2; ~6.0 Kb) has only a single ORF that encodes for the large (L) protein, also known as RNA-dependent RNA polymerase (RdRp).

Dichorhaviruses have been detected in multiple crops such as citrus (*Citrus* spp.), coffee (*Coffea* sp.), and several ornamentals including orchids [1]. *Dichorhavirus*-like OFV replicates inside mites [6], but plant-infected tissues develop localized chlorotic and/or necrotic lesions as the viruses do not move systemically [7]. The relationship between *Brevipalpus* spp. and *Dichorhavirus* transmissions is often very complex, as multiple false spider species are involved. OFV is the only *Brevipalpus*-transmitted virus (BTV) with worldwide distribution, transmitted by *B. californicus* to orchids, citrus, and ornamentals [8,9]. However, outside the family *Orchidaceae*, OFV has been found naturally infecting *Dieffenbachia* sp. (Araceae), *Swinglea glutinosa* (Rutaceae) [9], lilyturf (*Liriope spicata*, Asparagaceae) [10], green ti plant (*Cordyline terminalis*, Asparagaceae) [1], common hollyhock (*Alcea rosea*, Malvaceace) (https://hdl.handle.net/2263/84240, accessed on 18 February 2025), and spike speedwell (*Veronica spicata*, Plantaginaceae) [11]. Recent findings on rough lemon and mandarin orange (Rutaceae) in Hawaii [12], greenbrier (*Smilax auriculata*, Smilacaceae) [13], cast-iron plant (*Aspidistra elatior*), lilyturf or monkey grass (*Liriope muscari*), aztec grass (*Ophiopogon intermedius*, Asparagaceae) [14], and pandan grass (*Pandanus amaryllifolius*, Pandanaceae) in Florida [15], and in orchids (*Cymbidium* sp., *Dendrobium* sp., and *Dendrochilum magnum*) in California pose a potential threat to billion-dollar US citrus industry [16]. Current literature describes that OFV has two orchids (OFV-Orc1 and OFV-Orc2) and two citrus strains (OFV-Cit1 and OFV-Cit2). At least one of these four OFV-variants has been implicated in causing CiL disease (CiLD) in Colombia, Mexico, Hawaii, and South Africa [12,17,18,19,20,21], but there has been no report of OFV-Cit infection in orchids or other host species.

Several diagnostic methods have been developed for detecting viruses, but a limited number of diagnostic assays are available for *Dichorhavirus* detection [22,23,24,25,26,27]. Plant viruses are routinely detected with serological and molecular techniques such as the enzyme-linked immunosorbent assay (ELISA) and polymerase chain reaction (PCR), respectively. At present, no *Dichorhavirus* antibody is commercially available for serological tests. Only developed conventional RT-PCR assays are utilized for the routine detection of dichorhaviruses [22,23,24,25,26,27].

The e-Probe Diagnostic Nucleic Acid Analysis (EDNA) is a computational pipeline that utilizes multiple short (40–120 nt) pathogen-specific curated sequences, known as e-probes, to detect and identify known pathogens (single or multiple) of interest from raw HTS datasets [28]. To overcome the HTS challenges, like time-consuming and laborious data analysis and variation in laboratory-specific cutoff values for diagnostic decisions, the Microbe-Finder (MiFi^®^) online platform was created. The easy-to-use MiFi^®^ platform incorporates the EDNA pipeline to consolidate e-probe design and validation within a single interface [29]. There are two tools, MiProbe and MiDetect, incorporated in the MiFi^®^ web application to identify curated e-probes and detect/identify target organism/s from HTS libraries, respectively. This technology has been utilized earlier for the detection and identification of plum pox virus [30], citrus tristeza virus [31], hop viruses, and viroids [32] and has proven its capability in the detection of oomycetes [33], fungi [34], bacteria *Ralstonia solanacearum* [35], and fastidious prokaryotes like ‘*Candidatus* Liberibacter asiaticus’ and *Spiroplasma citri* [31]. Very recently, the MiFi^®^ metagenome analysis platform was used for the simultaneous detection of multiple pathogens associated with bovine respiratory disease (BRD) complex [36].

Here, we explored the EDNA technology integrated with the online MiFi^®^ platform for the detection of dichorhaviruses infecting multiple plant species, with emphasis on strain differentiation of OFV. In this study, e-probes for the genus *Dichorhavirus*, and specific e-probes for its six species, OFV strains, and its variants were developed. The detection accuracy of the EDNA technology was validated by comparison of results with the existing gold standard molecular diagnostics (RT-qPCR) and by detecting *Dichorhavirus* towards strain level using RNASeq data from multiple OFV-infected plant species. In the validation study, the sensitivity and specificity of each set of curated e-probes of *Dichorhavirus* were evaluated using HTS meta-transcriptomic libraries infecting citrus, orchid, and multiple ornamental plant species. Newly developed *Dichorhavirus* e-probes successfully identified *Farfugium japonicum* (family Asteraceae) as a new host for OFV and exposed the possibility of a new strain of OFV in nature.

## 2. Materials and Methods

### 2.1. Phylogenetic Analysis of Dichorhaviruses Using Available Genome Sequences in GenBank

Three lineages of dichorhaviruses identified in the phylogenetic trees based on L-protein (RNA2) were called sub-groups 1, 2, and 3 [23]. The topology of the phylogenetic tree that likely reflects a *Dichorhavirus* and mite vector coevolutionary relationship is very important to visualize the subgroups of dichorhaviruses before designing the generic and specific e-probes for disease diagnostics. To verify the phylogenetic relationship among dichorhaviruses, 28 complete *Dichorhavirus* RNA1 and RNA2 genome sequences available in GenBank were downloaded and aligned using the in-built MUSCLE program in MEGA 11 [37]. For this study, phylogenetic trees for RNA1 and RNA2 were created utilizing the Neighbor-Joining method [38] supported by 1000 bootstrap replicates (next to the branches) [39] and the evolutionary distances were computed using the Maximum Composite Likelihood method [40] (Figure 1).

### 2.2. Buildup of the Dichorhavirus e-Probe Sequences Infecting Different Hosts

To generate e-probes for leprosis-related dichorhaviruses, genomic data was retrieved from three sources: (1) GenBank and published reports, (2) In-house collection of BTVs (kitaviruses- and *Dichorhavirus*-infected/suspected samples sequences from Florida, California, Hawaii, Colombia, Costa Rica, and Mexico, and (3) Sequence data shared by the researchers working on BTVs. Primarily, scientists from the United States Department of Agriculture (USDA)–Agricultural Research Service, USDA–Animal Plant Health Inspection Service; Plant Protection Quarantine, and Institute for Biosecurity and Microbial Forensic at Oklahoma State University worked together and initiated e-probes development for dichorhaviruses infecting citrus, coffee, Clerodendrum, and orchid [16].

### 2.3. Development, Curation, and in Silico Validation of Dichorhavirus e-Probes

To generate e-probes, two specific FASTA formatted files are needed as input into EDNA: (1) a target genomes file and (2) a near neighbors’ file. The target and near neighbor files were retrieved using https://www.ncbi.nlm.nih.gov/taxonomy (accessed on 15 November 2022) website by entering the search term ‘*Dichorhavirus*’. As an example, to retrieve a set of target genomes for *Dichorhavirus leprosis* (CiLV-N) associated with CiLD in Brazil, the Taxonomy ID: 2560386 was retrieved and then the NCBI database (https://www.ncbi.nlm.nih.gov, accessed on 15 November 2022) was searched for the term “txid2560386 [Organism:exp]”. A custom sequence length in the range of 5500 to 7000 bp was used to capture the full-length sequences of RNA1 and RNA2 of dichorhaviruses. Then, the target sequences were downloaded in FASTA format, and the raw e-probes were developed and curated for BLASTn analysis. This file is the target genome needed for input into EDNA. To create the near neighbors’ file for e-probe design, the taxonomy ID: 1913605 was retrieved for the *Dichorhavirus* genus. To retrieve the near neighbor genomes from NCBI in FASTA format, the search term “(txid1913605 [Organism:exp]) NOT txid2560386 [Organism:exp]” was entered into https://www.ncbi.nlm.nih.gov/ (accessed on 15 November 2022) where the NOT txid Boolean statement excludes the target. Raw e-probe sequences that did not match the target pathogen (e-value of ≤1 × 10^−10^ were removed from the final e-probe set to ensure diagnostic specificity [28,30,41]. The files were uploaded into MiFi^®^, and e-probes for 40, 60, and 80 nucleotides were created using the e-probe developer software (MiProbe v2) inside MiFi^®^. These probes were further manually curated using BLASTn to the nr/nt database, excluding *Dichorhavirus leprosis* (CiLV-N), with Taxonomy ID: 2560386. E-probes that hit a different organism with a percent identity and query coverage greater or equal to 90% were removed. Furthermore, the newly developed e-probes were filtered to remove any known hits to the host genome, using the same percent identity and query coverage as that used for the nt database. The manually curated probes were then re-uploaded to MiFi^®^. For each target and near neighbor FASTA file, the same procedure was repeated for *Dichorhavirus citri* (CiCSV), *Dichorhavirus clerodendri* (ClCSV), *Dichorhavirus coffeae*, (CoRSV), *Dichorhavirus orchidaceae*, (OFV or citrus necrotic spot virus; CiNSV), and unclassified *Dichorhavirus* (CiBSV). In addition, the sequence of *Dichorhavirus* sp. ‘monocotyledonae’, an unclassified member, was added for validation of curated *Dichorhavirus* e-probes.

Pairwise genome sequence alignments for all RNA-1 and -2 genome segments were carried out using nucmer (v4.0.0rc1) [42] to establish links displaying similar regions among the studied dichorhaviruses. Gene annotations were retrieved from their respective accessions in the NCBI nucleotide database and e-probe target regions were identified between *Dichorhavirus* genus vs. species-specific. GenBank accessions of all 28 *Dichorhavirus* genomes were listed in the Appendix A. The Circos plot with corresponding links and annotated features was generated using Circos (v0.69-8) [43]. Labels and formatting were processed using Inkscape 1.3.2.

### 2.4. Detection of Dichorhaviruses Using Real-Time RT-qPCR Assays

There are four dichorhaviruses (OFV, CiLV-N, CiCSV, and CiBSV) that infect citrus and produce leprosis-like symptoms. Out of four dichorhaviruses, only OFV has four distinct variants (OFV-Orc1, OFV-Orc2, OFV-Cit1, and OFV-Cit2) which infect citrus [9,12,17,21]. To detect the dichorhaviruses (CiLV-N, CiCSV, and OFV) associated with citrus leprosis-like symptoms, a multiplex real-time RT-PCR assay was used [44]. For RNA-quality check, plant internal control nad5 gene primers and probes were also included in the assays. Furthermore, to differentiate the strain of OFVs in the infected tissue, separate real-time quadruplex RT-qPCR assays were optimized [45]. To validate the outcome of the e-probe analysis in the HTS data, RT-qPCR was utilized as a screening tool, whereas the bioinformatic analysis of meta-transcriptomic data [46] was used as a confirmatory diagnostic tool. RT-qPCR was performed in a QuantStudio^TM^ 5 Real-Time PCR System (Thermo Fisher Scientific Inc. Carlsbad, CA, USA) following the optimized manufacturer protocol [44,45].

### 2.5. High Throughput Sequencing of Dichorhaviruses from Different Hosts

For this study, an optimized Illumina ‘Ribo-Zero Total RNA’ HTS protocol [47] was utilized to extract total RNA from 31 samples consisting of 14 host species including *C. aurantium*, *C. reticulata*, *Sapium* sp., *Aralia* sp., orchids (*Cymbidium* sp., *Dendrobium* sp., and *Dendrochilum magnum*), and ornamentals (*Hibiscus rosa-sinensis*, *Liriope* sp., *Aspidistra* sp., *Ophiopogan* sp. *Smilax auriculata*, *Pedilanthus tithymaloides,* and *F. japonicum*) using the RNeasy Plant Mini Kit (Qiagen, Germantown, MD, USA) following a modification of the manufacturer protocol [46]. Before proceeding to the cDNA library preparation, the quality and concentration of extracted RNA were measured using a TapeStation (Agilent, Santa Clara, CA, USA). The Illumina TruSeq^R^ Stranded Total RNA Library Prep Plant kit (Illumina, Inc., San Diego, CA, USA) was utilized to construct the 31 cDNA libraries following the modified Illumina ‘Ribo-Zero Total RNA’ protocol [47]. Library titers were quantified using the Qubit fluorometer (Invitrogen, Carlsbad, CA, USA) and quality was tested using TapeStation (Agilent, Santa Clara, CA, USA) as per the manufacturer’s instructions. Either single- or pair-end sequencing was conducted using a MiSeq platform or NextSeq 550 system with 2 × 75 bp, Hi-output Illumina sequencing reagent kits (Illumina, San Diego, CA, USA). A published bioinformatic pipeline [46] was utilized to determine the genomic sequence of known and novel viruses/strains if any, in the generated FASTQ sequence data files.

### 2.6. Dichorhavirus Detection Using MiFi^®^ Platforms and Determination of the Diagnostic Sensitivity and Specificity of Its e-Probes

For the in vivo validation study, extracted total RNAs from 31 BTV suspected samples were evaluated before proceeding with HTS data generated by either MiSeq or Nextseq 550 Illumina platform. In total, 31 libraries (Table 1) were scanned using e-probes, and the outcome was compared with the real-time RT-qPCR results. For further confirmation, data was analyzed using an optimized bioinformatic pipeline [46]. The metagenome sequence output files (.fastq) were concatenated, compressed in gzip format, and uploaded into MiFi^®^ Metagenomes section. The developed 40, 60, and 80 nts probes were searched against all the libraries created for known and unassigned *Dichorhavirus* sp. using the MiDetect v2 software in the MiFi^®^ (https://bioinfo.okstate.edu, accessed on 14 March 2023) platform. To determine a positive alignment, an e-value ≤ 1 × 10^−10^ was used as the threshold. The analysis was based on comparing *t* test scores between the target (detection signal) and decoy (reverse e-probe sequence described as background signal) e-probes. True positives were called positive when the sequence library scanned using all three e-probes having different lengths (40, 60, and 80 nts) were predicted to be positive and matched with wet lab RT-qPCR assay to confirm that the sample contained the target pathogen. Likewise, true negatives were called negative when the sequence library scanned using e-probes (40, 60, and 80 nts) were predicted to be negative and matched laboratory confirmation (RT-qPCR-verified) that the sample was negative for the target pathogen. Contradictory suspicious, false negative and false positive e-probe diagnosis were further confirmed by bioinformatic analysis. The following formulae were used to determine the e-probe sensitivity (TP/TP + FN) and specificity (TN/TN + FP), where ‘TP’ = true positive, ‘TN’ = true negative, ‘FN’ = false negative, and ‘FP’ = false positive.

## 3. Results

### 3.1. e-Probe Development and Curation

A total of 28 *Dichorhavirus* RNA-1 and -2 genome segments sequences available in the GenBank, including 13 OFV (subgroup-1), 4 CiLV-N, 3 CiBSV (subgroup-2), 4 CiCSV, 3 ClCSV, and 1 CoRSV (subgroup-3), were included to generate the *Dichorhavirus* genus and species-specific e-probes (Figure 2). To determine the optimal size of *Dichorhavirus* e-probes, 40, 60, and 80 nt lengths of e-probes were designed (Table 1). The number of *Dichorhavirus* e-probe sequences was reduced after curation. In total, 684 *Dichorhavirus* e-probes were curated by comparing genome sequences against mono-segmented cytoplasmic and nuclear rhabdovirus genera assigned to the family *Rhabdoviridae*, with cileviruses belonging to the family *Kitaviridae* and other citrus infecting viruses. Furthermore, available host genome sequences of interest were also included in the BLASTn analysis to discard the sequences having identity with the host and retain the *Dichorhavirus*-specific e-probes of interest. All the *Dichorhavirus* e-probes were further categorized into the genus, species, strain, and variant-specific group. A total of 180 genus-specific; 394 species-specific (99, 114, and 181 e-probes for subgroup-1, -2, and -3, respectively); 10 OFV strain-specific (3 for OFV-Orc and 7 for OFV-Cit); 62 OFV-Orc (6 for Orc1 and 56 for Orc2); and 38 OFV-Cit (14 for Cit1 and 24 for Cit2) variant-specific e-probes were uploaded in the MiFi^®^ platform (Table 1). Except for OFV, the number of *Dichorhavirus* e-probes for each species was increased when the nucleotide length of the e-probe was reduced from 80 to 40. Overall, 40 mer size e-probes produced the maximum number, followed by 60 and 80 mer e-probes (Table 1). We have successfully designed 40 nt e-probes for all the *Dichorhavirus* species but failed to curate 60 and 80 nt strain-specific OFV-Cit, OFV-Orc e-probes, and 80 nts e-probe for OFV-Orc2 variant as there is a limited number of sequences for select species in the public databases (Table 1). Moreover, the smaller cutoff (40 mer e-probe) as compared to the larger cutoff (60 or 80 mer) has a higher potential to generate more e-probes for smaller pathogen genomes by the MiFi^®^ platform.

### 3.2. Screening of Potential Dichorhavirus-Infected Samples Using RT-qPCR Assays

A total of 31 BTV suspected samples belonging to 14 host species were collected from four countries (12 from USA, 12 from Mexico, 4 from Colombia, and 3 from Costa Rica) and tested for *Brevipalpus*-transmitted *Dichorhavirus* infection utilizing *Dichorhavirus* species-specific (OFV, CiCSV, CiLV-N) and OFV strain-specific RT-qPCR assays. There are no RT-qPCR diagnostic assays available for generic *Dichorhavirus*, CoRSV, CiBSV, and ClCSV, and generic OFV-Orc and OFV-Cit strain detection. Out of 31, only 16 OFV-positive samples were identified from Mexico (Querétaro, Colima, and Jalisco) and the USA (California, Florida, and Hawaii) using species-specific RT-qPCR assay. Among 16 samples, 13 were positive for either single or two OFV strains/variants. The positive samples included single infections of OFV-Cit1 (*n* = 1), OFV-Cit2 (*n* = 1), OFV-Orc2 (*n* = 5), and mixed infections of OFV-Orc1 and -Orc2 (*n* = 6) but the strains associated with the remaining three samples (*Liriope* S1-TH, *Aralia* CO002, and *Dendrobium* CO005ac) were undetermined (Table 2).

### 3.3. Detection of Dichorhavirus Genus, Species, OFV Strains, and Variants Utilizing e-Probes in HTS Data

To compare the results of RT-qPCR assays, all studied samples, including asymptomatic (healthy control), *Dichorhavirus* negative but symptomatic (negative control), and OFV positive (positive control), were processed for HTS, the meta-transcriptomic data uploaded into the Microbe Finder (MiFi^®^) platform and utilized for e-probe detection. The number of raw reads of RNASeq data generated by Illumina (MiSeq/NextSeq) sequencing platforms per sample varied from 9.5 to 89 million. MiFi detected *Dichorhavirus* in 15 RNASeq datasets using the e-probes designed for its species, strains, and variants whereas the remaining 16 samples were negative, even though one of the 16 samples (CO005ac) was suspicious for *Dichorhavirus* genus-specific e-probe diagnosis. Twelve out of fifteen were positive with all the generic *Dichorhavirus* e-probes sets (40, 60, and 80 nt) whereas the remaining three were positive with either 40 nt (FL7OHP) or 60 nt (CO002) length or both e-probes (ASP-FL) (Table 3). All the e-probe sets irrespective of length (40, 60, and 80 nts) successfully detected *Dichorhavirus* species OFV in 14 samples even though the total number of hits and the *p* values decreased as the e-probe length decreased (Table 3). Expected OFV strain-specific positive and unexpected false negative diagnostic calls were obtained for 12 and 2 OFV positive samples, respectively. Out of twelve, nine samples had a single infection with the OFV-Orc strain, two with the OFV-Cit strain, and one (sample CA2) with both strains. OFV strain-specific e-probes failed to clearly detect the presence of strains in the RNASeq data obtained from OFV-positive samples (FL7OHP and CA3) but successfully detected variant OFV-Orc1 or both the OFV-Orc variants in FL7OHP and CA3, respectively. Interestingly, the e-probe diagnosis was unsuccessful in determining the variant of OFV-Cit in the sample CA2. E-probe sets specific to OFV-Cit variants successfully detected OFV-Cit1 in the sample QR035 (*F. japonicum*) and OFV-Cit2 in JA022 (*C. aurantium)*. Expected negative diagnostic calls were obtained for 15 RT-qPCR negative samples, including the healthy control (JA023) and other 14 negative samples (Table 2) with the three e-probe sets, indicating that pathogen e-probes were specific to the *Dichorhavirus* species and OFV strains and not cross-reacting with the citrus, orchid, and other ornamental hosts genome sequences (Table 3). Interestingly, the sample CO002 (*Aralia* sp.) gave a positive diagnostic call for *Dichorhavirus* only with 60 nt length e-probe (*p*-value 0.0416). However, a suspicious call for OFV species raised the question for further confirmation. RNASeq data for BTV-suspected sample CO005ac (*Dendrobium* sp.) gave potentially true negative diagnostic calls with suspicious *p*-value 0.0794 for 40 nt *Dichorhavirus* generic e-probe only (Table 3).

### 3.4. Diagnostic Sensitivity and Specificity for Curated Dichorhavirus e-Probes in RNA-Seq Data and Confirmation of e-Probe Diagnostics Using Bioinformatic Pipeline Analysis

Diagnostic results of the *Dichorhavirus* e-probes from the field samples were compared with the results of *Dichorhavirus*-specific RT-qPCR assays to calculate diagnostic performance metrics using known positive, negative, and *Brevipalpus*-transmitted *Dichorhavirus*-suspected samples. In this study, 31 samples including citrus, orchids, and ornamentals were included for sensitivity and specificity testing and utilized metagenome compressed raw HTS fastq files from available RT-qPCR positive (*n* = 16) and negative (*n* = 15) samples uploaded to the MiFi^®^ platform. The EDNA diagnostic sensitivity and specificity analysis was calculated for curated e-probe sequences with 40-, 60-, and 80-nt e-probe lengths and evaluated for false-positive and -negative results (Table 4).

The number of raw reads RNASeq data ranged from 9.5 to 89 million but the total *Dichorhavirus* reads varies from minimum 372 [23.91% of 1556 total virus reads (0.007%), out of 21,801,326 post-trim reads] in FL7OHP to a maximum 10,766,019 [75.59% of 14,241,854 total virus reads (56.37%), out of 25,266,822 post-trim reads] in the sample S79_VOrBTP (Table 5). In contrast, maximum total virus reads (62,935,224/77,029,412 × 100 = 81.70%) were obtained in the sample S78_VDPCU with only 5.97% *Dichorhavirus* reads. Out of 22 *Dichorhavirus* positive samples, 11 had >90% *Dichorhavirus* reads with other viruses in mixed infection. Total virus reads from those 11 samples varied from 0.617% (119,926 reads) to 13.63% (3,013,247 reads) but *Dichorhavirus* reads represented between 91.49% and 99.94% of viral reads (Table 5). The relative abundance of *Dichorhavirus* sequence reads in the RNASeq data (Figure 3) was calculated using a curated *Dichorhavirus* e-probe irrespective of the optimal e-probe length (Table 5).

Among 16 RT-qPCR-positive samples, 15 samples were positive for *Dichorhavirus* e-probe. None of the *Dichorhavirus*-positive samples were identified as negative in e-probe diagnosis except the sample CO002. Interestingly, bioinformatic analysis revealed that the sample CO002 was infected with a single-segmented novel nucleo-rhabdovirus belonging to the family *Rhabdoviridae*. Furthermore, novel *Dichorhavirus* sequences were detected in six RNAseq libraries (sample CO005ac, TA018 from Mexico and S78_VDPCU, S79_VOrBTP, S80_VOrP, and S84_VBAP from Colombia) (Figure 3, Table 5) but none of the *Dichorhavirus* sequences in those six samples were detected by *Dichorhavirus* genus-specific e-probes except CO005ac which was diagnosed as suspicious (*p*-value 0.0794) diagnostic sample for possible *Dichorhavirus* infection. Out of 16 RT-qPCR-positive samples, OFV species-specific e-probes (OFV-Gen) were detected in 14 positive samples and diagnosed CO002 as suspicious and CO005ac as negative samples. OFV species-specific e-probe results show doubtful infection of OFV in the sample CO002 (*p*-value 0.0805) and negative in CO005ac (*p*-value 0.1195) as infected with a novel nucleo-rhabdovirus and a *Dichorhavirus*, respectively. In both cases, the OFV-specific e-probes were bioinformatically cross-reacted to the distantly related sequences, and EDNA sensitivity for OFV species-specific e-probe improved from 87.5% (14/16) to 100% after evaluating the results of RT-qPCR and EDNA with bioinformatic analysis. OFV strain-specific RT-qPCR detected two OFV-Cit and 11 OFV-Orc variants among 16 OFV-positive samples whereas it failed to determine the type of OFV strain as well as the type of variant present in the samples S1–TH (*Liriope* sp.), CO002 (*Aralia* sp.), and CO005ac (*Dendrobium* sp.) (Table 2). Even though e-probe analysis detected an OFV-Orc strain-specific sequence (*p*-value 0.0001) in the sample S1–TH, bioinformatic analysis confirmed the existence of an unrevealed OFV strain. Furthermore, the OFV-Orc1 variant was detected in CO002 by e-probe diagnostic (*p*-value 0.0209–0.0415), but the true positive e-probe diagnostic result turned into false positive as no similar e-probe sequence was identified by bioinformatic pipeline. The calculated sensitivity of the OFV variant-specific probe is 93.75% (15/16). The sensitivity of variant-level e-probe detection can be improved further by adding the new strain sequence in the strain-specific e-probe curation folder in the MiFi^®^ platform.

### 3.5. Discovery of a New Host Species of OFV and a Possible New OFV Strain in Known Hosts Utilizing e-Probe Diagnosis

Out of 14 host species studied for possible *Dichorhavirus* infection, 10 species (*C. aurantium*, *C. reticulata*, *Cymbidium* sp., *Dendrobium* sp., *Dendrochilum magnum*, *H. rosa-sinensis*, *Liriope* sp., *Aspidistra* sp., *Ophiopogan* sp., and *S. auriculata*) are known host for *Dichorhavirus*, whereas *P. tithymaloides*, *Sapium* sp., *Aralia* sp., and *F. japonicum* are not previously known as either natural or experimental hosts. Both preliminary RT-qPCR assays and the e-probe sets in the MiFi^®^ platform successfully detected *Dichorhavirus* and OFV in 12 (10 known and 2 unknown) host tissues from total RNA and 16 RNASeq datasets, respectively. The remaining fifteen samples, including one *P. tithymaloides* and two *Sapium* samples, and corresponding RNASeq datasets were negative by both methods. Both RT-qPCR and EDNA methods successfully detected OFV infection up to variant level (OFV-Cit1) in the new host *F. japonicum*, whereas e-probe diagnostic failed to identify any known OFV variant in two *Aspidistra* samples (ASP-FL and FL6ASP) from Tallahassee, Florida. On the other hand, OFV-specific RT-qPCR positive with C_T_ values 32.19 and 30.52 for Aralia (CO002) and Dendrobium (CO005ac) samples, respectively, were evaluated as false positive for *Dichorhavirus* infection by EDNA.

E-probe diagnostic results of dichorhaviruses’ presence or absence in symptomatic *F. japonicum* (QR035), *Aspidistra* sp. (ASP-FL and FL6ASP), *Aralia* sp. (CO002), and *Dendrobium* sp. (CO005ac) were validated using meta-transcriptomic data. During the bioinformatic analysis, the post-trim sequences were mapped to the Arabidopsis proteome and the available *Dendrobium* genome sequences in the NCBI database, and the identified host sequences were then removed. Out of 26,542,292 post-trims read in *F. japonicum* (QR035) cDNA library, 5,955,476 reads (23.44%) were mapped against plant viruses. The assembled contigs were blasted against the NCBI database and most of the total virus read matched with the genus *Nepovirus* (99.16%), *Ophiovirus* (0.54%), *Mycovirus* (0.23%), *Cytorhabdovirus* (0.045%), *Dichorhavirus* (0.0145%), and DNA plant virus sequences available in GenBank. In total, 11 *Dichorhavirus*-related contigs were obtained, with a maximum size of 2271 nt and a minimum of 197 nt. Assembled contigs cover 37.29% and 99.58% of RNA1 and RNA2 genome segments, respectively, and shared 99% nucleotide sequence identity with OFV-Cit1 (Acc. No. KF209275 and KF209276). Bioinformatics analysis confirmed the infection of OFV-Cit1 in *F. japonicum* obtained from El Pueblito in Querétaro, Mexico, and validated the e-probe diagnostic results.

Meta-transcriptomic analysis of two *Aspidistra* cDNA libraries (ASP-FL and FL6ASP) detected a total of 32,178 (0.037%) and 119,926 (0.62%) virus reads out of 87,154,532 and 19,424,805 post-trim reads, respectively. Furthermore, total virus read associated with ASP-FL and FL6ASP identified as DNA plant virus (8.11% and 0.46%), *Mycovirus* (89.96% and 0.72%), and *Dichorhavirus* (1.92% and 98.82%) read, respectively. In total, 11 *Dichorhavirus*-related contigs were obtained from ASP-FL RNASeq data, with a maximum size of 1245 nt and a minimum of 221 nt, whereas two large contigs (6027 and 6454) were obtained from FL6ASP RNASeq data. Assembled RNA1 and RNA2 genome contigs shared a maximum 90.63% and 98.43% nt identity with *Dichorhavirus orchidaceae* infected *V. spicata* (spike speedwell) in the United Kingdom (Acc. No. PP429912) and *P. amaryllifolius* (pandan grass) in Florida (OK624602), respectively. Data analysis confirmed the presence of a new OFV strain in the ornamental *Aspidistra* sp.

A total of 6,843,912 (31.12%) processed reads related to plant viruses were detected in the CO005ac library, which consists of mainly two virus genera: *Potexvirus* and *Dichorhavirus.* The 2nd highest percentage of viral reads (35.86%) were detected for *Dichorhavirus* after the *Potexvirus* reads (64.09%) (Table 2). The remaining 0.5% reads shared nucleotide identity either with *Badnavirus* or *Tobamovirus* sequences. The assembled contig (6827 nt) for *Potexvirus* was identified as cymbidium mosaic virus whereas the remaining two long contigs (6466 and 6053 nt) were identified as RNA 1 and RNA2 genome segment of an undescribed *Dichorhavirus*. BLASTn analysis of two contigs of *Dichorhavirus* (6466 and 6053 nt) genome segments shared 70.64% nt sequence identity with 34% genome coverage and 70.50% nt sequence identity with 67% genome coverage with OFV-Orc2 RNA1 (MW021482) and OFV-Cit2 RNA2 (MK578001), respectively, and identified as a novel species for *Dichorhavirus,* which is a distantly relative of OFV. Another novel *Dichorhavirus* species sequence was identified in the samples; S78_VDPCU, S79_VOrBTP, S80_VOrP, and S84_VBAP from Colombia via bioinformatic analysis but surprisingly none of the sample’s sequences computed any suspicious value for any generic or species-specific OFV e-probes.

Altogether 1,588,639 single-end Illumina reads recovered from the *Aralia* sp. (CO002) cDNA library were mapped to the *Rhabdovirus*, the highest proportion of viral reads (1,588,639/1,596,975 × 100 = 99.48%) in the sample, whereas the remaining 8336 viral reads were related to *Badnavirus* and endornavirus sequences (Table 4). Total *Rhabdovirus* reads were further categorized between two subfamilies, *Betarhabdovirinae* and *Alpharhabdovirinae* at the rate of 96.51% (1,533,153/1,588,639 × 100) and 3.49% (55,486/1,588,639 × 100), respectively. Two major contigs of 11,583 and 13,856 nt covering almost the entire single-segmented rhabdovirus genome were identified. The nucleotide sequence of contigs 1 and 2 shared 80.75% and 64% nt identities with query coverage of 98% and 8%, respectively, with the nucleotide sequences of datura yellow vein virus (DYVV, *Betanucleorhabdovirius*) (KM823531) infecting black-eyed Susan (*Thunbergia alata*) in Australia and eggplant mottled dwarf virus (EMDV) (OR613409) infecting eggplant (*Solanum melongena*) in Iran, respectively. Interestingly, no significant similarity was found in BLASTn analysis with the contig 1 query sequence and EMDV (*Alphanucleorhabdovirius*) genome sequence. However, the contig 1 sequence shared approximately 2% of genome coverage (1702 to 1801, 1956 to 1993, and 2378 to 2498 nts) and ~76% nt identity with OFV-Cit1 (KF209276), the member of the genus *Dichorhavirus.*

## 4. Discussion

*Dichorhavirus* infections mostly produce local chlorotic/necrotic lesions and/or chlorotic spots appearing on leaves and fruits in monocots as well as in dicots. The most economically important disease associated with *Dichorhavirus* infection is citrus leprosis, first described in Florida at the start of the 20th century [48], but its occurrence has not been observed since the mid-1960s. HTS of citrus-leprosis-like symptomatic herbarium samples from Florida revealed an association with a distant relative of OFV, referred to as CiLV-N0 [49]. The necrotic lesion symptoms observed in *C. sinensis* in Florida are associated with the *Dichorhavirus,* CiLV-N0 infection. Similar symptoms were observed in *Citrus* spp. in Colombia, Mexico, Hawaii, and South Africa but the identified pathogen in association with the leprosis-like symptom was OFV [9,12,17,18,19,21,50]. The economic consequences of *Dichorhavirus* infections, particularly CiLD, are substantial. The identification of new strains or hosts, as demonstrated in this study, could further complicate disease management strategies. The declining cost of HTS and the accessibility of computational biologists with advanced bioinformatics and computer programming knowledge have made it reasonable for many laboratories and plant regulatory agencies like USDA-APHIS-PPQ to implement this cutting-edge technology in diagnostics [47,51,52]. In the current research, we explored the EDNA technology [28] integrated with the online MiFi^®^ platform [29] to validate the sensitivity and specificity of curated *Dichorhavirus* e-probes by utilizing meta-transcriptomic libraries of BTVs belonging to the families *Kitaviridae* and *Rhabdoviridae*, along with other viruses infecting citrus and virus-free citrus and other hosts.

This study aimed to develop and curate e-probes at the genus, species, strain, and variant level using the *Dichorhavirus* species OFV as a model by adopting the scope of EDNA detection [53]. To adapt the approach according to the specific goal, generic e-probes at least for two or more *Dichorhavirus* species were designed utilizing the conserved regions of all the 28 *Dichorhavirus* genome sequences available in the GenBank. In contrast, non-conserved regions of the target genomes of multiple OFV isolates further identified at the strains/variants level were included to increase the sensitivity and specificity of e-probes and curated against phylogenetically related neighbors.

Previously, analytical performance metrics were assessed in silico based on a limit of detection of high-quality hits of e-probes [34], but the current study is focused on in vivo validation. HTS data generated from 31 RT-qPCR tested field samples collected from Colombia, Costa Rica, Mexico, and three different geographical locations of the United States were utilized to confirm the broader specificity of curated e-probes. Field samples collected from 14 different hosts, which have apparent *Brevipalpus* or eriophyid mite-transmitted virus symptoms were sequenced and analyzed using curated *Dichorhavirus* e-probes, and the test results were confirmed by comparing RT-qPCR and bioinformatic analysis. Both *Dichorhavirus* species-specific and OFV strain-specific RT-qPCR assays were developed in-house and validated for their sensitivity and specificity [44,45]. Similarly, the bioinformatic pipeline optimized for virus detection and discovery was validated in multiple studies [46,47]. To use the bioinformatic tools for pathogen detection, a person should have the capability to analyze the output of the Sequence Alignment Map (SAM) format before mapping the reference genome and/or BLAST to the GenBank database. MiFi^®^ eliminates the requirement of a dedicated bioinformatician as the user never creates any SAM or BLAST output files and provides a clear answer within 30 min.

Generally, the relative size of the pathogen and the number of raw e-probes are proportionally correlated as demonstrated for viroids, which are the smallest known pathogens and have the lowest number of e-probes [29,31]. To improve diagnostic specificity during *Dichorhavirus* e-probe curation, non-specific sequences were removed. In this study, we curated a total of 684 *Dichorhavirus* genus, species, strain, or variant-specific e-probes in the MiFi^®^ platform (Table 1). We showed that the EDNA can be utilized to detect the presence or absence of *Dichorhavirus*, the discovery of a new natural host (*F. japonicum*), and the possible existence of a novel *Rhabdovirus*/*Dichorhavirus* or a new strain of OFV in nature. Three sizes (40, 60, and 80 nt) of e-probes were successfully designed to match only the *Dichorhavirus* of interest, and variant-specific e-probes for OFV-Cit and OFV-Orc strains, except 80 nt e-probe for OFV-Orc2 variant. To assess the analytical sensitivity of curated e-probes, either in silico or in vitro validation is required to determine if the benchmark needs any further improvement [29]. In vitro validation of e-probe diagnostic must be accompanied by gold standard real-time PCR results of the same samples being sequenced. Here, we evaluated the sensitivity and specificity of all the genus, species, OFV strains, and its variant-specific e-probes utilizing EDNA incorporated in the MiFi^®^ platform. The performance of the MiFi^®^ platform was determined when EDNA results were compared and validated with the gold-standard RT-qPCR assays. In total, three cases of OFV species-specific RT-qPCR positive samples (*Liriope* sp. S1-TH, *Aralia* sp. CO002, and *Dendrobium* sp. CO005ac) failed to determine the type of variant by OFV strain-specific RT-qPCR assay (Table 2). Moreover, RT-qPCR assays were not conducted for generic *Dichorhavirus*, CoRSV, CiBSV, and ClCSV species, and generic OFV-Orc and OFV-Cit strain detection. Therefore, bioinformatic pipeline analysis was included as a 3rd method for further confirmation of e-probe diagnostic results. The RT-qPCR diagnostic was wrong in six cases when the e-probe diagnosis was verified by bioinformatic analysis. Out of 25 contradictory e-probe diagnoses, 17 were the correct diagnosis, whereas 8 were incorrect (Table 6). Among eight incorrect e-probe diagnoses, analysis results were either false positive (n = 5) or false negative (n = 3) as compared to RT-qPCR and bioinformatic analysis. Interestingly, two out of eight false diagnoses occurred in the samples CO002 and CO005ac, which were negative to listed dichorhaviruses in the study, but were later determined to have a novel nucleo-rhabdovirus in CO002 and a novel dichorhavirus in CO005ac present in RNASeq libraries. False-negative results when using e-probes are often caused by the inability of the *t*-test to compare the target with decoy e-probes and identify variance. This issue arises from either no hits or an excessive number of hits that reach the maximum limit of 250 hits permitted by EDNA. Typically, this can be addressed by adding internal control e-probes as used previously to calculate the variance with other citrus pathogens [31]. Internal control provides a small amount of background variance that assures that the statistical analysis does not have zero variance in the algorithm denominator, which results in a non-computed (NC) result. False positive detection of OFV-Orc1 in the sample CO002 indicated that the e-probe was cross-reacted with a novel nucleo-rhabdovirus sequence, and this is an expected behavior for e-probes since they are designed based on available data in public databases and are not meant to detect new virus species (Table 6). Overall, e-probe diagnosis incorrectly identified the OFV-Cit strain in the sample CA2 without identifying any variant, suggesting that the OFV strain-specific e-probes might not be 100% specific or bioinformatic analysis failed to pick up the low read OFV-Cit variant sequence in the CA2 RNASeq library. Moreover, the discovery of new OFV strain sequences in the samples *Liriope* sp. (S1-TH) and *Aspidistra* sp. (FL6ASP) RNASeq libraries also supports the RT-qPCR data and clarifies the reason behind the false detection of the presence or absence of different OFV-variant sequences through e-probe diagnosis (Table 6). E-probe synthesis is limited by the availability of target and near-neighbor genomic information and generally improves as more related sequences become available. Recently deposited four more OFV complete genome sequences (Acc. PP429909-10, PP429912-13, LC771578-79, and LC846649-50) and a new dichorhavirus species vinca chlorotic spot virus sequence (OR372158-59) in GenBank further extended the scope for more generic or specific dichorhavirus e-probe development.

Calculation of sequencing depth is an important parameter that can improve the chances of finding the desired pathogen reads in a complex metagenome, but no correlation was found between total viruses read and the ratio of *Dichorhavirus* and other virus read number in the studied RNASeq data (Figure 3, Table 5). The unbiased computer search will find the read if it is intact and present in the meta-transcriptomic data. At least one pathogen (OFV) was detected by MiFi^®^ in 14 out of 16 real-time PCR-positive (Ct values ≤ 34) samples. This indicates that the titers of OFV during an active disease process were high enough to accomplish metagenome-based e-probe detection. The relative abundance of *Dichorhavirus* reads is highest in the sample CA1 metagenome (0.13607), and lowest in the sample ASP-FL (7.09 × 10^−6^) (Table 5). The detection time depends on computer power, but if the sample is readily detected with curated e-probe in the MiFi^®^ platform, then hits will happen within minutes instead of 20–30 min required when pathogen reads are few among the total number of reads in the metagenome (Table 5). In conclusion, meta-transcriptomic-based *Dichorhavirus* detection with MiFi^®^ was successfully tested; most of the tested samples (29/31) were correctly diagnosed with minimum error and led to the discovery of a new nucleo-rhabdovirus and a novel *Dichorhavirus* species. Deposition of novel *Dichorhavirus* species sequences and additional OFV strain and variant sequences in the GenBank database will further improve the sensitivity and specificity of curated *Dichorhavirus* e-probes available at https://bioinfo.okstate.edu (accessed on 25 February 2025).

The advantages of the metagenomic-based bioinformatic approach to pathogen detection are further accelerated by utilizing the MiFi^®^-based sensitive, specific, and time-efficient e-probe diagnostic method application in the discovery of a novel host (*F. japonicum*). The broader implication of the MiFi^®^ is that the platform follows a rules-based approach (e-probe size, BLAST parameters, and scores) and more narrow search space to selecting the appropriate query sequence (e-probes) in an efficient manner since it eliminates irrelevant sequences that are later used to detect the pathogen in metagenomes compared to a traditional approach of screening metagenomes manually by using BLASTn directly against the NCBI’s more general databases such as nr/nt. Additionally, adjusting the e-value threshold and choosing “general” *Dichorhavirus* e-probes allowed for the search and discovery of a novel rhabdovirus and *Dichorhavirus* in meta-transcriptomic data that are not the specific target organism. Further addition of multiple OFV genome sequences from different hosts could improve the diagnostic rate for indistinct variants in the current database. Thus, the EDNA system [28] integrated into the MiFi^®^ platform [29] can be adjusted or designed to address a range of applications and/or scientific needs which will allow the research scientist to address multifaceted experiments where monitoring and surveillance, detection, and diagnosis of pathogens infecting plant, animal, and human are required. The potential limitation of the MiFi^®^ platform and e-probe curation is its dependency on sequences from the public domain, such as the near-neighbor sequence selection. It could take an expert to weed out any spurious sequences. Additionally, it takes a specialist to understand the current and usually esoteric classification system of the organism and create the probes in the proper context. For example, a person should have a thorough knowledge of taxonomy to create probes for viruses specific to the genus, species, strain, or variant. It would also be interesting to observe how e-probe diagnosis performs between the HTS data generated from the same cDNA library utilizing higher and lower error rates of the nanopore and Illumina sequencing platforms.

## 5. Conclusions

In this study, we explored millions of reads of cDNA sequences generated from 31 suspected plant samples for their ability to match with the newly curated short sequence/s (e-probe) specific to *Dichorhavirus* genus, species, OFV strains, and variants. A platform (MiFi^®^) integrated with multiple complex software was utilized to detect the targeted short-specific *Dichorhavirus* sequence from the tested dataset. Newly developed e-probes for dichorhaviruses correctly diagnosed 29 tested samples with minimum error and led to the discovery of a new host (leopard plant, *F. japonicum*), new ornamental OFV strain, and novel rhabdoviruses. The results of the EDNA method were validated with the gold standard RT-qPCR diagnostic methods and confirmed by bioinformatic pipeline analysis. Overall, the EDNA method is very specific and sensitive and can detect the targeted pathogen within one to 30 min, depending on the total number of pathogens reads in the sequencing dataset. The same protocol can be used by other scientists to develop, curate, and validate e-probes related to any virus genera and in the discovery of new hosts, new virus species, new strains, or variants. The findings of this study further pave the way for future investigations into the *Dichorhavirus* research. A focus on the evolutionary dynamics of *Dichorhavirus* would aid in prediction and help stakeholders manage their potential impact on agriculture and natural ecosystems. Implementation of the EDNA method and updated host information for *Dichorhavirus* diagnosis should assist regulatory agencies in surveillance activities to monitor the distribution pattern of citrus leprosis-associated dichorhaviruses in alternate hosts in countries where it is present and to prevent dissemination into citrus growing countries where there is no report of these viruses.

## Figures and Tables

**Figure 1 viruses-17-00441-f001:**
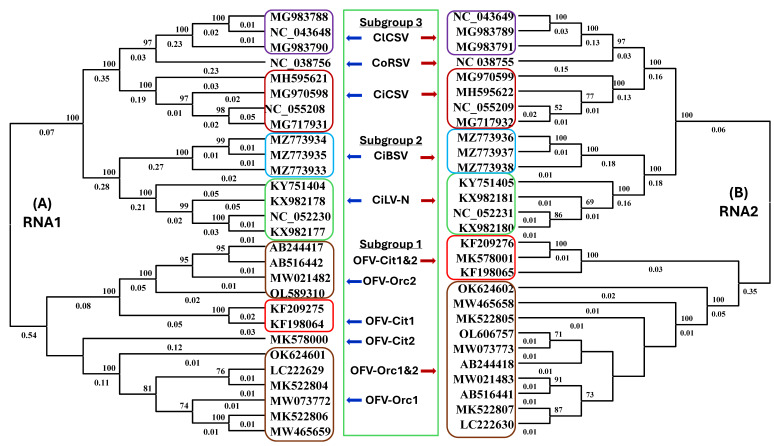
Phylogenetic relationships of dichorhaviruses [orchid fleck virus (OFV), citrus leprosis virus N (CiLV-N), citrus chlorotic spot virus (CiCSV), citrus bright spot virus (CiBSV), clerodendrum chlorotic spot virus (ClCSV), and coffee ring spot virus (CoRSV)] using neighbor-joining methods based on (A) RNA1 and (B) RNA2 genome sequences. Phylogenies were supported by 1000 bootstrap replicates and bootstrap values greater than 50 are shown at the nodes. Genetic pieces of information associated with the NCBI accessions mentioned in the distinct subgroup clades were considered for e-probes design.

**Figure 2 viruses-17-00441-f002:**
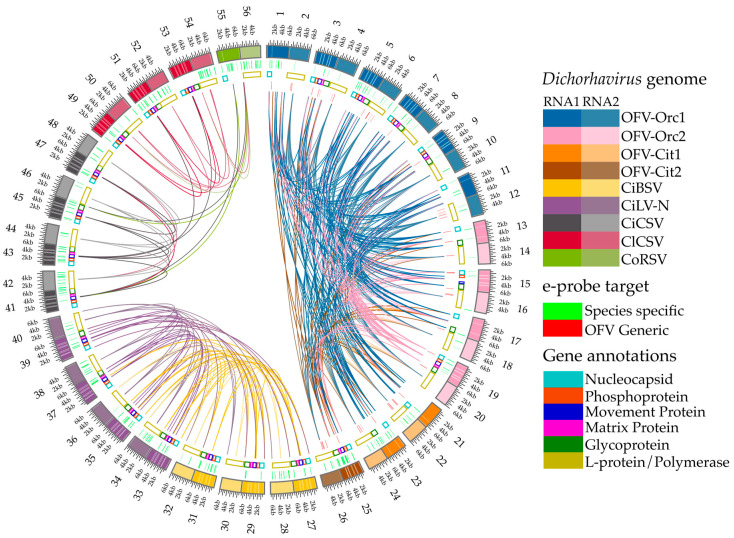
Multiple pairwise whole genome alignment with gene annotations and highlights for e-probe target regions of the dichorhaviruses used in this study. Alignments performed with the MiProbe v2 software revealed regions of genomic similarity between the targeted dichorhaviruses, and the links between viruses display local high-identity pairwise alignments. The number showing outside the outermost ring represents three subgroups of dichorhavirus; 1–26 represent subgroup 1 (OFV), 27–40 represent subgroup 2 (CiBSV and CiLV-N), and 41–56 represent subgroup 3 (CiCSV, ClCSV, and CoRSV). Odd and even numbers outside the circle represent *Dichorhavirus* RNA1 and RNA2 genome segments, respectively. From the outermost to innermost rings: (outermost ring) nucleotide positions of RNA1 and RNA2 of genomes of dichorhaviruses; (inner ring 1) green highlights corresponding to e-probes targeting exclusively specific virus species; (inner ring 2) gene annotations retrieved from NCBI reports for all viruses for both RNA1 and RNA2 segments; (innermost half-circle) red highlights corresponding to e-probes designed for orchid fleck virus (OFV) generic regions shared by multiple targets. As stated by the latest NCBI annotations of the studied accessions, hypothetical proteins and orphan ORFs were not considered for gene annotation display.

**Figure 3 viruses-17-00441-f003:**
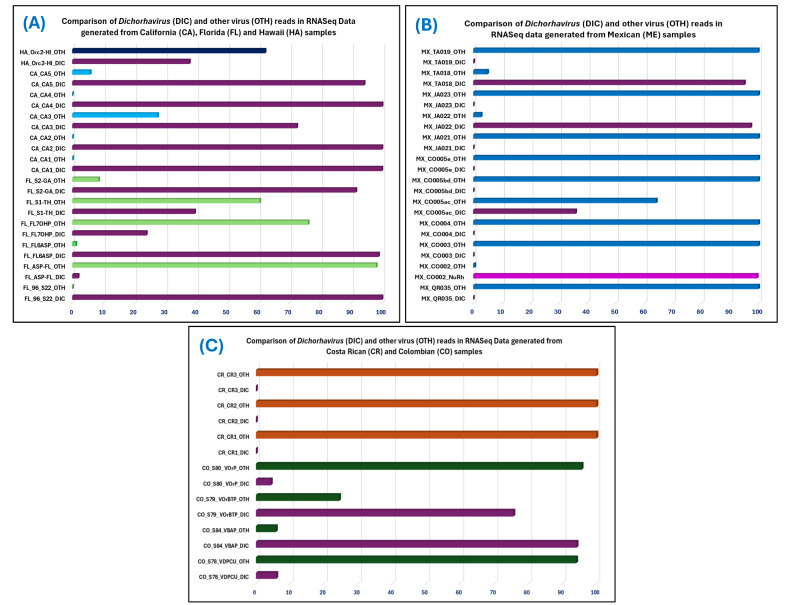
Distribution of rhabdovirus and other virus reads (OTH) among total virus reads detected in 31 RNASeq data, which were obtained from *Brevipalpus*-transmitted virus-suspected samples from four countries: (**A**) USA, (**B**) Mexico (**C**) Colombia and Costa Rica. Samples from each country are represented by individual colors in the bar diagrams. Single-segment nucleo-rhabdovirus (NuRh) reads in the samples are represented by purple, bi-segment *Dichorhavirus* (DIC) is represented by plum color, whereas other viruses are represented by green (Florida), light blue (California), dark blue (Hawaii), in the USA samples, blue color in Mexican samples, dark green in Colombia samples, and orange color in Costa Rica samples.

**Table 1 viruses-17-00441-t001:** Three different lengths (40, 60, and 80 nt) of *Dichorhavirus* genus, species, strain, and variant-specific e-probes were designed using the e-probe developer software (MiProbe v2) inside the Microbe Finder (MiFi^®^) platform.

Group Based on Phylogenetic Relationship	Virus Genus/ Species/Strain /Variant	Number of Available Sequence in GenBank	e-Probes					
			Length (nt)	Genome Segment		Sub-Total	Total	Grand Total
				**RNA1**	**RNA2**			
	*Dichorhavirus*	28	40	35	32	67	180	684
			60	34	24	58		
			80	34	22	56		
Subgroup 1	Orchid fleck virus (OFV) Generic	13	40	6	22	28	99	
			60	14	20	34		
			80	16	21	37		
	Orchid fleck virus citrus strain (OFV-Cit)	3	40	1	6	7	7	
			60	0	0	0		
			80	0	0	0		
	Orchid fleck virus citrus strain, variant 1 (OFV-Cit1)	2	40	4	0	4	14	
			60	5	0	5		
			80	5	0	5		
	Orchid fleck virus citrus strain, variant 2 (OFV-Cit2)	1	40	8	0	8	24	
			60	10	0	10		
			80	6	0	6		
	Orchid fleck virus orchid strain (OFV-Orc)	10	40	1	2	3	3	
			60	0	0	0		
			80	0	0	0		
	Orchid fleck virus orchid strain, variant 1 (OFV-Orc1)	6	40	22	0	22	56	
			60	19	0	19		
			80	15	0	15		
	Orchid fleck virus orchid strain, variant 2 (OFV-Orc2)	4	40	5	0	5	6	
			60	1	0	1		
			80	0	0	0		
Subgroup 2	Citrus bright spot virus (CiBSV)	3	40	13	8	21	36	
			60	7	2	9		
			80	5	1	6		
	Citrus leprosis virus N (CiLV-N)	4	40	15	16	31	78	
			60	16	9	25		
			80	13	9	22		
Subgroup 3	Citrus chlorotic spot virus (CiCSV)	4	40	16	13	29	73	
			60	11	15	26		
			80	8	10	18		
	Clerodendrum chlorotic spot virus (ClCSV)	3	40	10	13	23	65	
			60	9	13	22		
			80	8	12	20		
	Coffee ring spot virus (CoRSV)	1	40	10	8	18	43	
			60	7	8	15		
			80	4	6	10		

**Table 2 viruses-17-00441-t002:** Detection of *Dichorhavirus* species [orchid fleck virus (OFV), citrus leprosis virus N (CiLV-N), and citrus chlorotic spot virus (CiCSV)] and four variants of orchid fleck virus (OFV-Orc1, OFV-Orc2, OFV-Cit1, and OFV-Cit2) infecting citrus, orchids, and ornamentals in the USA, Mexico, Colombia, and Costa Rica using two separate Taqman RT-qPCR assays. No RT-qPCR diagnostic assays are available for citrus bright spot virus (CiBSV), clerodendrum chlorotic spot virus (ClCSV), coffee ring spot virus (CoRSV), and generic orchid (OFV-Orc) and citrus (OFV-Cit) strains of OFV. * Refers to the absence of RT-qPCR assay. NT refers to ‘not tested’ and UD refers to ‘undetermined’.

	Name				RT-qPCR Diagnostic Based on C_T_ Value													Comments Based on C_T_ Value Obtained by RT-qPCR
					Genus Specific	Species Specific						OFV Strain Specific						
Sl. No.	Host	Isolate	City, State	Country	*Dichorhavirus* *	CiBSV *	CiCSV	CiLV-N	ClCSV	CoRSV *	OFV-Gen	OFV-Cit *	OFV-Cit1	OFV-Cit2	OFV-Orc *	OFV-Orc1	OFV-Orc2	
1	*Smilax auriculata*	96_S22	Gainesville, Florida	USA	NT	NT	UD	UD	UD	NT	13.92	NT	UD	UD	NT	27.74	15.66	Positive for OFV-Gen, OFV-Orc1 & OFV-Orc2
2	*Liriope* sp.	S2-GA			NT	NT	UD	UD	UD	NT	14.35	NT	UD	UD	NT	UD	18.52	Positive for OFV-Gen, & OFV-Orc2
3		S1-TH	Tallahassee, Florida		NT	NT	UD	UD	UD	NT	13.87	NT	UD	UD	NT	UD	UD	Positive for OFV-Gen
4	*Aspidistra* sp.	ASP-FL			NT	NT	UD	UD	UD	NT	23.47	NT	UD	UD	NT	UD	33.26	Positive for OFV-Gen, & OFV-Orc2
5		FL6ASP			NT	NT	UD	UD	UD	NT	17.42	NT	UD	UD	NT	31.56	27.71	Positive for OFV-Gen, OFV-Orc1 & OFV-Orc2
6	*Ophiopogan* sp.	FL7OHP			NT	NT	UD	UD	UD	NT	26.39	NT	UD	UD	NT	31.37	29.12	Positive for OFV-Gen, OFV-Orc1 & OFV-Orc2
7	*Cymbidium* sp.	CA1	San Diego, California		NT	NT	UD	UD	UD	NT	18.05	NT	UD	UD	NT	UD	17.08	Positive for OFV-Gen, & OFV-Orc2
8		CA4			NT	NT	UD	UD	UD	NT	18.38	NT	UD	UD	NT	UD	18.97	Positive for OFV-Gen, & OFV-Orc2
9	*Dendrobium* sp.	CA2			NT	NT	UD	UD	UD	NT	14.03	NT	UD	UD	NT	15.91	18.66	Positive for OFV-Gen, OFV-Orc1 & OFV-Orc2
10		CA3			NT	NT	UD	UD	UD	NT	22.26	NT	UD	UD	NT	26.05	22.99	Positive for OFV-Gen, OFV-Orc1 & OFV-Orc2
11	*Dendrochilum magnum*	CA5			NT	NT	UD	UD	UD	NT	14.48	NT	UD	UD	NT	30.44	14.56	Positive for OFV-Gen, OFV-Orc1 & OFV-Orc2
12	*Citrus reticulata*	Orc2-HI	Hilo, Hawaii		NT	NT	UD	UD	UD	NT	23.67	NT	UD	UD	NT	UD	25.94	Positive for OFV-Gen, & OFV-Orc2
13	*Farfugium japonicum*	QR035	El Pueblito, Querétaro	Mexico	NT	NT	UD	UD	UD	NT	24.32	NT	27.92	UD	NT	UD	UD	Positive for OFV-Gen, & OFV-Cit1
14	*Aralia* sp.	CO002	Manzanillo, Colima,		NT	NT	UD	UD	UD	NT	32.19	NT	UD	UD	NT	UD	UD	Positive for OFV-Gen
15	*Hibiscus rosa-sinensis*	CO003			NT	NT	UD	UD	UD	NT	UD	NT	UD	UD	NT	UD	UD	Negative for tested dichorhavirus species
16		CO004			NT	NT	UD	UD	UD	NT	UD	NT	UD	UD	NT	UD	UD	Negative for tested dichorhavirus species
17	*Dendrobium* sp.	CO005ac			NT	NT	UD	UD	UD	NT	30.52	NT	UD	UD	NT	UD	UD	Positive for OFV-Gen
18		CO005bd			NT	NT	UD	UD	UD	NT	UD	NT	UD	UD	NT	UD	UD	Negative for tested dichorhavirus species
19		CO005e			NT	NT	UD	UD	UD	NT	UD	NT	UD	UD	NT	UD	UD	Negative for tested dichorhavirus species
20	*Citrus aurantium*	JA021	Sayula, Jalisco		NT	NT	UD	UD	UD	NT	UD	NT	UD	UD	NT	UD	UD	Negative for tested dichorhavirus species
21		JA022			NT	NT	UD	UD	UD	NT	13.44	NT	UD	17.99	NT	UD	UD	Positive for OFV-Gen, & OFV-Cit2
22		JA023	Zapotlanejo, Jalisco		NT	NT	UD	UD	UD	NT	UD	NT	UD	UD	NT	UD	UD	Negative for tested dichorhavirus species
23	*Hibiscus rosa-sinensis*	TA018	Villahermosa, Tabasco		NT	NT	UD	UD	UD	NT	UD	NT	UD	UD	NT	UD	UD	Negative for tested dichorhavirus species
24	*Pedilanthus tithymaloides*	TA019			NT	NT	UD	UD	UD	NT	UD	NT	UD	UD	NT	UD	UD	Negative for tested dichorhavirus species
25	Orchids	S78_VDPCU	Pradera, Valle del Cauca	Colombia	NT	NT	UD	UD	UD	NT	UD	NT	UD	UD	NT	UD	UD	Negative for tested dichorhavirus species
26		S79_ VOrBTP			NT	NT	UD	UD	UD	NT	UD	NT	UD	UD	NT	UD	UD	Negative for tested dichorhavirus species
27		S80_VOrP			NT	NT	UD	UD	UD	NT	UD	NT	UD	UD	NT	UD	UD	Negative for tested dichorhavirus species
28		S84_VBAP			NT	NT	UD	UD	UD	NT	UD	NT	UD	UD	NT	UD	UD	Negative for tested dichorhavirus species
29	*Liriope* sp.	CR1	San José	Costa Rica	NT	NT	UD	UD	UD	NT	UD	NT	UD	UD	NT	UD	UD	Negative for tested dichorhavirus species
30	*Sapium* sp.	CR2			NT	NT	UD	UD	UD	NT	UD	NT	UD	UD	NT	UD	UD	Negative for tested dichorhavirus species
31		CR3			NT	NT	UD	UD	UD	NT	UD	NT	UD	UD	NT	UD	UD	Negative for tested dichorhavirus species

**Table 3 viruses-17-00441-t003:** Detection of *Dichorhavirus* in RNASeq data originated from healthy and *Brevipalpus* transmitted virus suspected symptomatic leaf tissues utilizing e-probe sets designed for *Dichorhavirus* species [coffee ring spot virus (CoRSV), citrus leprosis virus N (CiLV-N), citrus chlorotic spot virus (CiCSV), clerodendrum chlorotic spot virus (ClCSV), citrus bright spot virus (CiBSV), and orchid fleck virus (OFV)], OFV strains (OFV-Orc and OFV-Cit), and variants (OFV-Orc1, OFV-Orc2, OFV-Cit1, and OFV-Cit2) available in the MiFi platform. Out of 31, 16 RT-qPCR positive, 1 healthy and 1 negative sample e-probe analysis data were included. Except for OFV, none of the *Dichorhavirus* species was identified in the listed samples, and negative results for CiLV-N, CiCSV, CoRSV, ClCSV, and CiBSV are not included in this table. The target and decoy scores were compared using a *t*-test. Three tiers of diagnostic calls were used in the statistical test, positive (*p*-value ≤ 0.05), suspect (0.05 > *p*-value ≤ 0.1), and negative (*p*-value > 0.1). No significant difference between the two sets (target and decoy) indicated no evidence for the presence of pathogen sequences, and the sample was designated negative for the *Dichorhavirus*. The negative and suspicious e-probe *p*-values were highlighted using orange and lavender colors, respectively. Blue shaded ‘NP’ refers to ‘no probe’ while shadeless ‘NC’ refers to ‘not computed’.

RNASeq Data Generated from Different Host (Isolate)	Genus, Species Strain and Variant Specific e-Probes	Pairwise *t* Test Diagnostic Based on Type of Libraries and Legth of e-Probes								EDNA Diagnotic Results Based on Combined Analysis of 40, 60 and 80 NTs e-Probes
		**Single Library**				**Concatenated Libraries**				
		***p* Value**			**Diagnostic**	***p* Value**			**Diagnostic**	
		**40 nt**	**60 nt**	**80 nt**		**40 nt**	**60 nt**	**80 nt**		
*Smilax auriculata* (96_S22)	Dichorhavirus	1.95 × 10^−5^	0.0002	0.0025	Positive	1.58 × 10^−5^	0.0001	0.0005	Positive	True Positive for Dichorhavirus, OFV-Gen, OFV-Orc, OFV-Orc1 & OFV-Orc2, suspicious for OFV-Cit
	OFV-Gen	6.93 × 10^−12^	6.93 × 10^−10^	3.52 × 10^−6^	Positive	5.32 × 10^−12^	7.11 × 10^−10^	2.53 × 10^−8^	Positive	
	OFV-Cit	0.1029	NP	NP	Negative	0.0735	NP	NP	Suspicious	
	OFV-Cit1	0.2119	NC	NC	Negative	0.2119	NC	NC	Negative	
	OFV-Cit2	NC	NC	NC	Negative	NC	NC	NC	Negative	
	OFV-Orc	2.32 × 10^−5^	NP	NP	Positive	3.46 × 10^−5^	NP	NP	Positive	
	OFV-Orc1	0.0411	0.0209	NC	Positive	0.0023	0.0004	0.1753	Positive	
	OFV-Orc2	1.71 × 10^−7^	0.5	NP	Positive	1.95 × 10^−7^	0.5	NP	Positive	
*Aspidistra* sp. (Asp-FL)	Dichorhavirus	0.0209	0.0596	0.1135	Positive	0.0006	0.0099	0.0591	Positive	True Positive for Dichorhavirus, OFV-Gen, & OFV-Orc suspicious for OFV-Orc1
	OFV-Gen	0.0002	0.0007	0.0031	Positive	4.37 × 10^−5^	0.0004	0.0003	Positive	
	OFV-Cit	NC	NP	NP	Negative	NC	NP	NP	Negative	
	OFV-Cit1	NC	NC	NC	Negative	NC	NC	NC	Negative	
	OFV-Cit2	NC	NC	NC	Negative	NC	NC	NC	Negative	
	OFV-Orc	0.0921	NP	NP	Negative	0.0380	NP	NP	Positive	
	OFV-Orc1	0.0811	NC	NC	Suspicious	0.0946	NC	NC	Suspicious	
	OFV-Orc2	NC	NC	NP	Negative	NC	NC	NP	Negative	
*Aspidistra* sp. (FL6ASP)	Dichorhavirus	0.0006	0.0103	0.0287	Positive	6.64 × 10^−5^	0.0022	0.0131	Positive	True Positive for Dichorhavirus, OFV-Gen, & OFV-Orc, suspicious for OFV-Cit, OFV-Orc1 and OFV-Cit2
	OFV-Gen	6.48 × 10^−8^	5.71 × 10^−6^	5.94 × 10^−5^	Positive	3.43 × 10^−9^	2.36 × 10^−6^	4.55 × 10^−5^	Positive	
	OFV-Cit	0.1957	NP	NP	Negative	0.0485	NP	NP	Suspicious	
	OFV-Cit1	0.2119	NC	NC	Negative	0.2119	NC	NC	Negative	
	OFV-Cit2	NC	0.1231	0.2023	Negative	0.0853	0.1158	NC	Suspicious	
	OFV-Orc	0.0921	NP	NP	Suspicious	0.0380	NP	NP	Positive	
	OFV-Orc1	0.0811	NC	NC	Suspicious	0.0946	NC	NC	Negative	
	OFV-Orc2	NC	NC	NP	Negative	NC	NC	NP	Negative	
*Ophiopogan* sp. (FL7OHP)	Dichorhavirus	0.0049	0.0902	0.1094	Positive	0.0011	0.0589	0.0873	Positive	True Positive for Dichorhavirus, OFV-Gen, & OFV-Orc1, suspicious for OFV-Orc
	OFV-Gen	7.85 × 10^−5^	0.0008	0.0115	Positive	6.60 × 10^−7^	6.88 × 10^−5^	0.0037	Positive	
	OFV-Cit	NC	NP	NP	Negative	NC	NP	NP	Negative	
	OFV-Cit1	NC	NC	NC	Negative	NC	NC	NC	Negative	
	OFV-Cit2	NC	NC	NC	Negative	NC	NC	NC	Negative	
	OFV-Orc	0.0938	NP	NP	Suspicious	0.0938	NP	NP	Suspicious	
	OFV-Orc1	0.0415	0.1717	NC	Positive	0.0076	0.1753	0.0814	Positive	
	OFV-Orc2	0.2119	NC	NP	Negative	0.2119	NC	NP	Negative	
*Liriope* sp. (S1-TH)	Dichorhavirus	0.0010	0.0063	0.0288	Positive	5.30 × 10^−7^	5.04 × 10^−6^	3.88 × 10^−6^	Positive	True Positive for Dichorhavirus, OFV-Gen, OFV-Orc, OFV-Orc1, & OFV-Orc2, suspicious for OFV-Cit
	OFV-Gen	3.33 × 10^−7^	4.54 × 10^−5^	0.0001	Positive	1.76 × 10^−12^	2.45× 10^−8^	9.89 × 10^−9^	Positive	
	OFV-Cit	NC	NP	NP	Negative	0.0593	NP	NP	Suspicious	
	OFV-Cit1	NC	NC	NC	Negative	0.2119	NC	NC	Negative	
	OFV-Cit2	NC	0.1299	NC	Negative	0.1305	0.2023	0.2023	Negative	
	OFV-Orc	0.0606	NP	NP	Suspicious	0.0001	NP	NP	Positive	
	OFV-Orc1	0.0058	0.0266	0.0663	Positive	2.58 × 10^−5^	0.0104	0.0412	Positive	
	OFV-Orc2	0.0319	NC	NP	Positive	3.53 × 10^−8^	NC	NP	Positive	
*Liriope* sp. (S2-GA)	Dichorhavirus	0.0010	0.0032	0.0003	Positive	4.15 × 10^−6^	6.60 × 10^−5^	0.0003	Positive	True Positive for Dichorhavirus, OFV-Gen, OFV-Cit, OFV-Orc & OFV-Orc2
	OFV-Gen	3.09 × 10^−6^	1.95 × 10^−5^	0.0003	Positive	2.21 × 10^−13^	3.58 × 10^−9^	2.41 × 10^−8^	Positive	
	OFV-Cit	0.1957	NP	NP	Negative	0.0422	NP	NP	Positive	
	OFV-Cit1	NC	NC	NC	Negative	0.2119	NC	NC	Negative	
	OFV-Cit2	NC	NC	NC	Negative	NC	NC	NC	Negative	
	OFV-Orc	0.0654	NP	NP	Suspicious	3.49 × 10^−5^	NP	NP	Positive	
	OFV-Orc1	NC	NC	NC	Negative	NC	0.1717	NC	Negative	
	OFV-Orc2	0.0192	NC	NP	Positive	1.14 × 10^−7^	0.5	NP	Positive	
*Cymbidium* sp. (CA1)	Dichorhavirus	3.17 × 10^−5^	0.0004	0.0044	Positive	1.98 × 10^−5^	0.0002	0.0028	Positive	True Positive for Dichorhavirus, OFV-Gen, OFV-Orc, & OFV-Orc2
	OFV-Gen	3.79 × 10^−12^	6.47 × 10^−10^	5.37 × 10^−8^	Positive	6.75 × 10^−13^	5.46 × 10^−10^	1.09 × 10^−8^	Positive	
	OFV-Cit	0.1957	NP	NP	Negative	0.1957	NP	NP	Negative	
	OFV-Cit1	NC	NC	NC	Negative	NC	NC	NC	Negative	
	OFV-Cit2	NC	NC	NC	Negative	NC	NC	NC	Negative	
	OFV-Orc	0.0001	NP	NP	Positive	0.0001	NP	NP	Positive	
	OFV-Orc1	NC	NC	NC	Negative	NC	NC	NC	Negative	
	OFV-Orc2	4.50 × 10^−8^	NC	NP	Positive	3.40 × 10^−8^	0.5	NP	Positive	
*Dendrobium* sp. (CA2)	Dichorhavirus	4.40 × 10^−6^	4.89 × 10^−5^	0.0004	Positive	9.79 × 10^−7^	1.17 × 10^−5^	1.04 × 10^−5^	Positive	True Positive for Dichorhavirus, OFV-Gen, OFV-Cit, OFV-Orc, OFV-Orc1 & OFV-Orc2 and suspicious for OFV-Cit2
	OFV-Gen	1.26 × 10^−14^	2.24 × 10^−9^	4.93 × 10^−7^	Positive	9.49 × 10^−17^	6.51 × 10^−12^	2.28 × 10^−8^	Positive	
	OFV-Cit	0.0432	NP	NP	Positive	0.0586	NP	NP	Suspicious	
	OFV-Cit1	NC	NC	NC	Negative	0.2119	NC	NC	Negative	
	OFV-Cit2	NC	0.1841	NC	Negative	0.1908	0.0854	0.2023	Suspicious	
	OFV-Orc	0.0317	NP	NP	Positive	3.55 × 10^−5^	NP	NP	Positive	
	OFV-Orc1	1.51 × 10^−20^	3.06 × 10^−11^	5.41 × 10^−5^	Positive	5.52 × 10^−40^	7.85 × 10^−32^	1.58 × 10^−6^	Positive	
	OFV-Orc2	2.53 × 10^−3^	NC	NP	Positive	0.0029	NC	NP	Positive	
*Dendrobium* sp. (CA3)	Dichorhavirus	9.66 × 10^−5^	2.17 × 10^−8^	0.0236	Positive	0.0001	0.0012	0.0131	Positive	True Positive for Dichorhavirus, OFV-Gen, OFV-Orc1 & OFV-Orc2 and suspicious for OFV-Orc
	OFV-Gen	5.38 × 10^−8^	2.26 × 10^−7^	6.19 × 10^−5^	Positive	7.01 × 10^−8^	8.84 × 10^−8^	1.62 × 10^−5^	Positive	
	OFV-Cit	NC	NP	NP	Negative	NC	NP	NP	Negative	
	OFV-Cit1	NC	NC	NC	Negative	NC	NC	NC	Negative	
	OFV-Cit2	NC	NC	NC	Negative	NC	NC	NC	Negative	
	OFV-Orc	0.0938	NP	NP	Suspicious	0.0820	NP	NP	Suspicious	
	OFV-Orc1	8.11 × 10^−2^	1.72 × 10^−1^	NC	Positive	5.73 × 10^−5^	1.55 × 10^−6^	0.0096	Positive	
	OFV-Orc2	9.73 × 10^−3^	NC	NP	Positive	0.0081	NC	NP	Positive	
*Cymbidium* sp. (CA4)	Dichorhavirus	3.51 × 10^−5^	0.0005	0.0067	Positive	2.77 × 10^−5^	0.0003	0.0033	Positive	True Positive for Dichorhavirus, OFV-Gen, OFV-Orc & OFV-Orc2
	OFV-Gen	5.84 × 10^−12^	8.31 × 10^−10^	1.05 × 10^−7^	Positive	2.61 × 10^−12^	5.52 × 10^−10^	1.74 × 10^−8^	Positive	
	OFV-Cit	0.1957	NP	NP	Negative	0.1957	NP	NP	Negative	
	OFV-Cit1	NC	NC	NC	Negative	NC	NC	NC	Negative	
	OFV-Cit2	NC	NC	NC	Negative	NC	NC	NC	Negative	
	OFV-Orc	0.0008	NP	NP	Positive	0.0002	NP	NP	Positive	
	OFV-Orc1	NC	NC	NC	Negative	NC	NC	NC	Negative	
	OFV-Orc2	3.53 × 10^−8^	NC	NP	Positive	3.44 × 10^−8^	NC	NP	Positive	
*Dendrochilum magnum* (CA5)	Dichorhavirus	3.35 × 10^−5^	0.0004	0.0060	Positive	2.58 × 10^−5^	0.0003	0.0030	Positive	True Positive for Dichorhavirus, OFV-Gen, OFV-Orc, OFV-Orc1 & OFV-Orc2
	OFV-Gen	6.85 × 10^−12^	7.11 × 10^−10^	1.07 × 10^−7^	Positive	3.38 × 10^−12^	5.39 × 10^−10^	2.29 × 10^−8^	Positive	
	OFV-Cit	0.1957	NP	NP	Negative	0.1957	NP	NP	Negative	
	OFV-Cit1	NC	NC	NC	Negative	NC	NC	NC	Negative	
	OFV-Cit2	NC	NC	NC	Negative	NC	NC	NC	Negative	
	OFV-Orc	0.0004	NP	NP	Positive	0.0001	NP	NP	Positive	
	OFV-Orc1	0.1699	NC	NC	Negative	0.0107	NC	NC	Positive	
	OFV-Orc2	3.48 × 10^−8^	0.5	NP	Positive	3.44 × 10^−8^	0.5	NP	Positive	
*Citrus reticulata* (Orc2-HI)	Dichorhavirus	0.0416	0.1629	0.1630	Positive	1.67 × 10^−5^	0.0001	0.0008	Positive	True Positive for Dichorhavirus, OFV-Gen, OFV-Orc, & OFV-Orc2
	OFV-Gen	0.0217	NC	NC	Positive	1.96 × 10^−11^	7.14 × 10^−9^	6.37 × 10^−8^	Positive	
	OFV-Cit	NC	NP	NP	Negative	NC	NP	NP	Negative	
	OFV-Cit1	NC	NC	NC	Negative	NC	NC	NC	Negative	
	OFV-Cit2	NC	NC	NC	Negative	NC	NC	NC	Negative	
	OFV-Orc	NC	NP	NP	Negative	0.0001	NP	NP	Positive	
	OFV-Orc1	NC	NC	NC	Negative	NC	0.1717	NC	Negative	
	OFV-Orc2	NC	NC	NP	Negative	8.88 × 10^−8^	0.5	NP	Positive	
*Aralia* sp. (CO002)	Dichorhavirus	0.0794	0.0416	NC	Positive	0.0794	0.0416	NC	Positive	True Positive for Dichorhavirus and suspicious for OFV-Gen
	OFV-Gen	NC	NC	NC	Negative	0.0805	0.0802	NC	Suspicious	
	OFV-Cit	NC	NP	NP	Negative	NC	NP	NP	Negative	
	OFV-Cit1	NC	NC	NC	Negative	NC	NC	NC	Negative	
	OFV-Cit2	NC	NC	NC	Negative	NC	NC	NC	Negative	
	OFV-Orc	NC	NP	NP	Negative	NC	NP	NP	Negative	
	OFV-Orc1	NC	NC	NC	Negative	0.0415	0.0209	NC	Positive	
	OFV-Orc2	NC	NC	NP	Negative	NC	NC	NP	Negative	
*Dendrobium* sp. (CO005ac)	Dichorhavirus	0.0794	NC	NC	Suspicious	0.0794	NC	NC	Suspicious	Suspicious for dichorhavirus as infected with Novel dichorhavirus
	OFV-Gen	0.1153	NC	NC	Negative	0.1195	NC	NC	Negative	
	OFV-Cit	NC	NP	NP	Negative	NC	NP	NP	Negative	
	OFV-Cit1	NC	NC	NC	Negative	NC	NC	NC	Negative	
	OFV-Cit2	NC	NC	NC	Negative	NC	NC	NC	Negative	
	OFV-Orc	NC	NP	NP	Negative	NC	NP	NP	Negative	
	OFV-Orc1	NC	NC	NC	Negative	NC	NC	NC	Negative	
	OFV-Orc2	NC	NC	NP	Negative	NC	NC	NP	Negative	
*Farfugium japonicum* (QR035)	Dichorhavirus	0.0137	0.0536	NC	Positive	0.0013	0.0139	0.0488	Positive	True Positive for Dichorhavirus, OFV-Gen, OFV-Cit, & OFV-Cit1
	OFV-Gen	0.0013	0.0005	0.0173	Positive	3.93 × 10^−6^	1.85 × 10^−5^	7.61 × 10^−5^	Positive	
	OFV-Cit	0.0239	NP	NP	Positive	0.0046	NP	NP	Positive	
	OFV-Cit1	0.1021	0.0892	NC	Suspicious	0.0256	0.0434	0.2119	Positive	
	OFV-Cit2	NC	NC	NC	Negative	0.1908	NC	NC	Negative	
	OFV-Orc	NC	NP	NP	Negative	0.2525	NP	NP	Negative	
	OFV-Orc1	NC	NC	NC	Negative	NC	NC	NC	Negative	
	OFV-Orc2	NC	NC	NP	Negative	NC	NC	NP	Negative	
*Citrus aurantium* (JA022)	Dichorhavirus	0.0002	0.0011	1.89 × 10^−2^	Positive	2.23 × 10^−5^	0.0001	0.0038	Positive	True Positive for Dichorhavirus, OFV-Gen, OFV-Cit, & OFV-Cit2 and Suspicious for OFV-Orc2
	OFV-Gen	7.48 × 10^−8^	1.39 × 10^−6^	4.65 × 10^−5^	Positive	8.22 × 10^−9^	2.66 × 10^−7^	2.20 × 10^−6^	Positive	
	OFV-Cit	6.85 × 10^−8^	NP	NP	Positive	1.23 × 10^−9^	NP	NP	Positive	
	OFV-Cit1	NC	NC	NC	Negative	NC	NC	NC	Negative	
	OFV-Cit2	0.0001	7.76 × 10^−6^	0.0042	Positive	2.26 × 10^−6^	3.89 × 10^−14^	0.0014	Positive	
	OFV-Orc	0.2525	NP	NP	Negative	0.1962	NP	NP	Negative	
	OFV-Orc1	0.1584	0.1717	NC	Negative	0.1548	0.1717	NC	Negative	
	OFV-Orc2	NC	NC	NP	Negative	0.0892	NC	NP	Suspicious	
*Citrus aurantium* (JA023)	Dichorhavirus	NC	NC	NC	Negative	NC	NC	NC	Negative	True Negative for all the e-probes
	OFV-Gen	NC	NC	NC	Negative	NC	NC	NC	Negative	
	OFV-Cit	NC	NP	NP	Negative	NC	NP	NP	Negative	
	OFV-Cit1	NC	NC	NC	Negative	NC	NC	NC	Negative	
	OFV-Cit2	NC	NC	NC	Negative	NC	NC	NC	Negative	
	OFV-Orc	NC	NP	NP	Negative	NC	NP	NP	Negative	
	OFV-Orc1	NC	NC	NC	Negative	NC	NC	NC	Negative	
	OFV-Orc2	NC	NC	NP	Negative	NC	NC	NP	Negative	
*Sapium* sp. (CR2)	Dichorhavirus	NC	NC	NC	Negative	NC	NC	NC	Negative	True Negative for all the e-probes as infected with Emravirus
	OFV-Gen	NC	NC	NC	Negative	NC	NC	NC	Negative	
	OFV-Cit	NC	NP	NP	Negative	NC	NP	NP	Negative	
	OFV-Cit1	NC	NC	NC	Negative	NC	NC	NC	Negative	
	OFV-Cit2	NC	NC	NC	Negative	NC	NC	NC	Negative	
	OFV-Orc	NC	NP	NP	Negative	NC	NP	NP	Negative	
	OFV-Orc1	NC	NC	NC	Negative	NC	NC	NC	Negative	
	OFV-Orc2	NC	NC	NP	Negative	NC	NC	NP	Negative	

**Table 4 viruses-17-00441-t004:** In vivo sensitivity and specificity testing of *Dichorhavirus* e-probes comparing the gold standard RT-qPCR data and bioinformatic pipeline analysis. ‘ND’ means ‘Not Determined’ and ‘NT’ means ‘Not Tested’.

**True Positive (TP) Detection by EDNA** **Diagnosis Using e-Probes of Dif** **ferent Length**							**False Negative (FN) Detection by EDNA Diagnosis in Comparison with RT-qPCR and Bioinformatic Analysis**						**Sensitivity (TP/TP** **+** **FN) of EDNA Diagnostics in Comparison with RT-qPCR and Bioinformatic Analysis**						**Validation of False Negative (FN) Detected in EDNA Diagnosis by Comparing the Results of**		**Sensitivity (TP/TP** **+** **FN) of EDNA** **Diagnosis in** **Comparison with**
**40 nt**		**60 nt**		**80 nt**		**All Sizes Together**	**40 nt**		**60 nt**		**80 nt**		**40 nt**		**60 nt**		**80 nt**		**RT-qPCR**	**RT-qPCR and/or** **Bioinformatic** **Analysis**	**RT-qPCR and** **Bioinformatic Analysis Plus** **RT-qPCR**
**Single**	**Concate**	**Single**	**Concate**	**Single**	**Concate**	**Combined**	**Single**	**Concate**	**Single**	**Concate**	**Single**	**Concate**	**Single**	**Concate**	**Single**	**Concate**	**Single**	**Concate**			
14	14	11	14	10	12	15	0	0	4	1	4	2	1	1	0.7895	0.9375	0.7895	0.8824	ND	1	ND/0.9375
14	14	13	14	13	14	14	0	0	1	0	1	0	1	1	0.9333	1	0.9333	1	1	0	0.9333/1
3	3	ND	ND	ND	ND	3	0	0	ND	ND	ND	ND	1	1	NT	NT	NT	NT	ND	0	ND/1
1	1	1	1	1	1	1	0	0	0	0	0	0	1	1	1	1	1	1	1	0	0.5/1
0	1	0	1	0	0	1	1	0	1	0	1	0	0.5	1	0.5	1	0.5	1	1	0	0.5/1
5	10	ND	ND	ND	ND	10	5	2	ND	ND	ND	ND	0.6667	0.8333	NT	NT	NT	NT	ND	1	ND/0.9091
5	6	4	5	1	4	6	1	0	1	0	1	0	0.8571	1	0.8571	1	0.8571	1	1	0	0.8571/1
8	9	0	0	ND	ND	9	2	1	9	9	ND	ND	0.8182	0.9	0.5	0.5	NT	NT	4	1	0.6923/0.9
**True Negative (TN) Detection by EDNA Diagnosis Using e-Probes of Different Length**							**False Positive (FP) Detection by EDNA** **Diagnosis in Comparison with RT-qPCR and Bioinformatic Analysis**						**Specificity (TN/TN** **+** **FP) of EDNA** **Diagnostics in Comparison with RT-qPCR and Bioinformatic Analysis**						**Validation of False Positive (FP) Detected in EDNA** **Diagnosis by Comparing the Results of**		**Specificity (TN/TN** **+** **FP) of EDNA** **Diagnosis in** **Comparison with**
**40 nt**		**60 nt**		**80 nt**		**All Sizes Together**	**40 nt**		**60 nt**		**80 nt**		**40 nt**		**60 nt**		**80 nt**		**RT-qPCR**	**RT-qPCR and/or** **Bioinformatic** **Analysis**	**RT-qPCR and** **Bioinformatic** **Analysis Plus RT-qPCR**
**Single**	**Concate**	**Single**	**Concate**	**Single**	**Concate**	**Combined**	**Single**	**Concate**	**Single**	**Concate**	**Single**	**Concate**	**Single**	**Concate**	**Single**	**Concate**	**Single**	**Concate**			
17	17	20	17	21	19	16	0	0	1	1	0	0	1	1	0.941176	0.941176	1	1	ND	1	ND/0.9412
17	17	18	17	18	17	17	0	0	0	0	0	0	1	1	1	1	1	1	0	0	Both are 1
28	28	ND	ND	ND	ND	28	1	2	ND	ND	ND	ND	0.9655	0.9333	NT	NT	NT	NT	ND	1	ND/0.9655
30	30	30	30	30	30	30	0	0	0	0	0	0	1	1	1	1	1	1	0	0	Both are 1
31	30	30	30	31	31	30	0	0	0	0	0	0	1	1	1	1	1	1	0	0	Both are 1
26	21	ND	ND	ND	ND	21	0	1	ND	ND	ND	ND	1	0.9546	NT	NT	NT	NT	ND	0	ND/1
26	25	27	26	30	27	25	4	6	3	4	1	1	0.8621	0.8065	8929	0.862069	0.9615	0.9615	2	2	Both are 0.9259
23	22	31	31	ND	ND	22	1	1	0	0	ND	ND	0.9565	0.9565	1	1	NT	NT	1	1	Both are 0.9565

**Table 5 viruses-17-00441-t005:** In vivo validation of *Dichorhavirus* e-probe diagnostic in 31 RNASeq data by comparing RT-qPCR results and bioinformatic pipeline analysis.

Sl. No.	Isolate	Country	Meta-Transcriptomic Analysis								Sensitivity	Specificity		
			Read					Total Contig Lentgh Matched with *Dichorhavirus*/ Nucleorhabdovirus	Total Virus Read (%)	*Dichorhavirus*/ Nucleorhabdovirus Among Total Virus Read (%)	Relative Abundance (RA) of Pathogen in the Sample	Detection of *Dichorhavirus*/Nucleorhabdovirus Related Sequences Using Bioinformatic Analysis of RNASeq Data	EDNA Diagnosis	**Comments Based on C_T_ Value Obtained by RT-qPCR**
			**Raw**	**Post Trim**	**Total Virus**	***Dichorhavirus*/Nucleorhabdovirus**	**Other Virus**							
1	96_S22	USA	29,026,369	28,668,880	893,873	893,236	637	12,510	3.11	99.93	0.03116	OFV Srtain-Orc2	True Positive for Dichorhavirus, OFV-Gen, OFV-Orc, OFV-Orc1 & OFV-Orc2, suspicious for OFV-Cit	Positive for OFV-Gen, OFV-Orc1 & OFV-Orc2
2	ASP-FL		89,070,401	87,154,532	32,178	618	31,560	2040	0.04	1.92	7.09 × 10^−6^	New OFV Srtain OFV-Orc3	True Positive for Dichorhavirus, OFV-Gen, & OFV-Orc suspicious for OFV-Orc1	Positive for OFV-Gen, & OFV-Orc2
3	FL6ASP		19,575,807	19,424,805	119,926	118,513	1413	12,841	0.62	98.82	0.00610	OFV-Orc2 & New OFV Srtain OFV-Orc3	True Positive for Dichorhavirus, OFV-Gen, & OFV-Orc, suspicious for OFV-Cit, OFV-Orc1 and OFV-Cit2	Positive for OFV-Gen, OFV-Orc1 & OFV-Orc2
4	FL7OHP		21,964,465	21,801,326	1556	372	1184	4707	0.01	23.91	1.71 × 10^−5^	New OFV Srtain OFV-Orc3	True Positive for Dichorhavirus, OFV-Gen, & OFV-Orc1, suspicious for OFV-Orc	Positive for OFV-Gen, OFV-Orc1 & OFV-Orc2
5	S1-TH		122,785,064	121,388,722	7,716,120	3,047,882	4,668,238	18,378	6.36	39.50	0.02511	New OFV Srtain OFV-Orc3	True Positive for Dichorhavirus, OFV-Gen, OFV-Orc, OFV-Orc1, & OFV-Orc2, suspicious for OFV-Cit	Positive for OFV-Gen
6	S2-GA		91,745,716	90,743,284	6,963,254	6,370,557	592,697	12,463	7.67	91.49	0.07020	OFV Srtain-Orc2	True Positive for Dichorhavirus, OFV-Gen, OFV-Cit, OFV-Orc & OFV-Orc2	Positive for OFV-Gen, & OFV-Orc2
7	CA1		22,261,865	22,116,296	3,013,247	3,009,450	3797	12,479	13.63	99.87	0.13607	OFV Srtain-Orc2	True Positive for Dichorhavirus, OFV-Gen, OFV-Orc, & OFV-Orc2	Positive for OFV-Gen, & OFV-Orc2
8	CA2		20,763,435	20,598,618	1,269,615	1,268,336	1279	18,662	6.16	99.90	0.06157	OFV Srtain-Orc1 & Orc2	True Positive for Dichorhavirus, OFV-Gen, OFV-Cit, OFV-Orc, OFV-Orc1 & OFV-Orc2 and suspicious for OFV-Cit2	Positive for OFV-Gen, OFV-Orc1 & OFV-Orc2
9	CA3		23,475,312	23,312,790	13,451	9742	3709	14,933	0.06	72.43	0.00042	OFV Srtain-Orc1 & Orc2	True Positive for Dichorhavirus, OFV-Gen, OFV-Orc1 & OFV-Orc2 and suspicious for OFV-Orc	Positive for OFV-Gen, OFV-Orc1 & OFV-Orc2
10	CA4		19,604,543	19,458,011	1,476,793	1,475,932	861	12,506	7.59	99.94	0.07585	OFV Srtain-Orc2	True Positive for Dichorhavirus, OFV-Gen, OFV-Orc & OFV-Orc2	Positive for OFV-Gen, & OFV-Orc2
11	CA5		24,331,027	24,173,029	1,832,898	1,725,961	106,937	12,752	7.58	94.17	0.07140	OFV Srtain-Orc2	True Positive for Dichorhavirus, OFV-Gen, OFV-Orc, OFV-Orc1 & OFV-Orc2	Positive for OFV-Gen, OFV-Orc1 & OFV-Orc2
12	Orc2-HI		50,962,963	48,290,861	6,721,283	2,540,314	4,180,969	12,467	13.92	37.80	0.05260	OFV Srtain-Orc2	True Positive for Dichorhavirus, OFV-Gen, OFV-Orc, & OFV-Orc2	Positive for OFV-Gen, & OFV-Orc2
13	QR035	Mexico	26,721,386	26,542,292	5,955,476	862	5,954,614	8367	3.60	0.01	3.25 × 10^−5^	OFV Srtain-Cit1	True Positive for Dichorhavirus, OFV-Gen, OFV-Cit, & OFV-Cit1	Positive for OFV-Gen, & OFV-Cit1
14	CO002		6,947,727	6,879,156	1,596,975	1,588,639	8336	25,439	23.22	99.48	0.23094	Novel nucleo rhabdovirus	True Positive for Dichorhavirus and suspicious for OFV-Gen	Positive for OFV-Gen
15	CO003		17,110,470	16,983,502	14,635,242	0	14,635,242	0	86.17	0.00	0	Negative to any rhabdovirus	True Negative for all the e-probes	Negative for tested dichorhavirus species
16	CO004		17,427,101	17,286,898	4358	0	4358	0	0.03	0.00	0	Negative to any rhabdovirus	True Negative for all the e-probes	Negative for tested dichorhavirus species
17	CO005ac		22,162,878	21,989,748	6,843,912	2,454,149	4,389,763	12,766	31.12	35.86	0.11160	Novel dichorhavirus	Suspicious for dichorhavirus as infected with Novel dichorhavirus	Positive for OFV-Gen
18	CO005bd		23,278,875	23,102,829	9,897,923	0	9,897,923	0	42.84	0.00	0	Negative to any rhabdovirus	True Negative for all the e-probes	Negative for tested dichorhavirus species
19	CO005e		24,703,710	24,520,564	7,404,784	0	7,404,784	0	30.20	0.00	0	Negative to any rhabdovirus	True Negative for all the e-probes	Negative for tested dichorhavirus species
20	JA021		20,346,774	20,193,394	1338	0	1338	0	0.01	0.00	0	No virus detected	True Negative for all the e-probes	Negative for tested dichorhavirus species
21	JA022		20,685,498	20,523,285	263,176	255,960	7216	18,481	1.28	97.26	0.01247	OFV Srtain-Cit2	True Positive for Dichorhavirus, OFV-Gen, OFV-Cit, & OFV-Cit2 and Suspicious for OFV-Orc2	Positive for OFV-Gen, & OFV-Cit2
22	JA023		17,124,047	16,929,790	147	0	147	0	0.00	0.00	0	No virus (Healthy like)	True Negative for all the e-probes	Negative for tested dichorhavirus species
23	TA018		15,660,539	15,459,953	269,342	255,846	13,496	12,551	17.38	94.99	0.01655	Novel dichorhavirus	True Negative for all the e-probes	Negative for tested dichorhavirus species
24	TA019		22,257,306	22,092,419	574,354	768	573,586	10,532	2.60	0.13	3.48 × 10^−5^	Negative to any rhabdovirus	True Negative for all the e-probes	Negative for tested dichorhavirus species
25	S78_VDPCU	Colombia	78,148,248	77,029,412	62,935,224	3,754,190	59,181,034	12,962	81.70	5.97	0.04874	Novel dichorhavirus	True Negative for all the e-probes	Negative for tested dichorhavirus species
26	S79_VOrBTP		26,126,253	25,266,822	14,241,854	10,766,019	3,475,835	12,515	56.37	75.59	0.42609	Novel dichorhavirus	True Negative for all the e-probes	Negative for tested dichorhavirus species
27	S80_VOrP		38,962,137	38,028,459	108,237	4724	103,513	12,460	0.28	4.37	0.00012	Novel dichorhavirus	True Negative for all the e-probes	Negative for tested dichorhavirus species
28	S84_VBAP		80,430,198	79,341,663	1,760,691	1,657,883	102,808	12,510	2.22	94.16	0.02090	Novel dichorhavirus	True Negative for all the e-probes	Negative for tested dichorhavirus species
29	CR1	Costa Rica	31,309,832	31,105,043	14,052	0	14,052	0	0.05	0.00	0	Negative to any rhabdovirus	True Negative for all the e-probes	Negative for tested dichorhavirus species
30	CR2		26,277,825	26,098,518	368,003	0	368,003	0	1.41	0.00	0	Negative to any rhabdovirus	True Negative for all the e-probes	Negative for tested dichorhavirus species
31	CR3		9,538,604	9,374,241	72,214	0	72,214	0	0.77	0.00	0	Negative to any rhabdovirus	True Negative for all the e-probes	Negative for tested dichorhavirus species

**Table 6 viruses-17-00441-t006:** Validation of contradictory results of *Dichorhavirus* e-probe diagnostic utilizing confirmatory bioinformatic pipeline analysis. ‘UD’ refers to ‘undetermined’, which means that in all the cycles of the RT-qPCR reaction the signal from the target cDNA did not pass the threshold level. ‘NT’ refers to ‘not tested’ due to absence of RT-qPCR assay.

Sl. No	Sample	*Dichorhavirus*	e-Probe Size Used in EDNA Detection (nt)			Method Used for Analysis		EDNA Diagnostic as Compare to RT-qPCR Only				Changed in RT-qPCR Detection after Bioinformatic Analysis	EDNA Diagnostic as Compare to RT-qPCR and Bioinformatic Analysis				EDNA Diagnosis	Overall Comment on e-Probe Specificity Based on Analysis of Atleast Two Methods
		Genus/ Species/Strain/ Variant						True Positive (TP)	True Negative (TN)	False Positive (FP)	False Negative (FN)		True Positive (TP)	True Negative (TN)	False Positive (FP)	False Negative (FN)		
			40	60	80	RT-qPCR (C_T_ Value)	**Bioinformatic Pipeline**											
1	ASP-FL (*Aspidistra* sp.)	OFV_Orc2	NEG	NEG	No probe	33.26	NEG				1	TP to TN		1			Negative	Correct
2	FL6ASP (*Aspidistra* sp.)	OFV_Orc1	NEG	NEG	NEG	31.56	NEG				1	TP to TN		1			Negative	Correct
		OFV-Orc	POS	No probe	No probe	NT	POS	Inconsistent (IC)				IC to TP	1				Positive	Correct
		OFV_Orc2	NEG	NEG	No probe	27.71	POS				1	TP to TP (No change)				1	False negative	Incorrect
3	FL7OHP (*Ophiopogan* sp.)	Dichorhavirus	POS	NEG	NEG	NT	POS	Inconsistent				IC to TP	1				Positive	Correct
		OFV_Orc1	POS	NEG	NEG	31.37	NEG	1				TP to TN	1				Positive	Correct
		OFV_Orc2	NEG	NEG	No probe	29.12	NEG				1	TP to TN		1			Negative	Correct
4	S1 (Liriope-TH)	OFV_Orc1	POS	POS	POS	UD	NEG			1		TN to TN (No change)			1		False positive	Incorrect
		OFV_Orc2	POS	NEG	No probe	UD	NEG			1		TN to TN (No change)			1		False positive	Incorrect
5	S2 (Liriope-GA)	OFV_Orc2	POS	NEG	No probe	18.52	POS	1				TP to TP (No change)	1				Positive	Correct
6	OFV-Hawaii (*Citrus reticulata*)	OFV_Orc2	POS	NEG	NEG	25.94	POS	1				TP to TP (No change)	1				Positive	Correct
7	CA1 (*Cymbidium* sp.)	OFV_Orc2	POS	NEG	NEG	17.08	POS	1				TP to TP (No change)	1				Positive	Correct
8	CA2 (*Dendrobium* sp.)	OFV-Cit	POS	No probe	No probe	NT	NEG	Inconsistent				IC to TN			1		False positive	Incorrect
		OFV_Orc2	NEG	NEG	No probe	18.66	POS				1	TP to TP (No change)	1				Positive	Correct
9	CA3 (Dendrobium sp.)	OFV-Orc	NEG	No probe	No probe	NT	POS	Inconsistent				IC to TP				1	False negative	Incorrect
		OFV_Orc2	POS	NEG	No probe	22.99	POS	1				TP to TP (No change)	1				Positive	Correct
10	CA4 (Cymbidium sp.)	OFV_Orc2	POS	NEG	No probe	18.97	POS	1				TP to TP (No change)	1				Positive	Correct
11	CA5 (*Dendrochilum magnum*)	OFV_Orc1	POS	NEG	NEG	30.44	NEG	1				TP to TP (No change)	1				Positive	Correct
		OFV_Orc2	POS	NEG	No probe	14.56	POS	1				TP to TP (No change)	1				Positive	Correct
12	CO002 (*Aralia* sp.)	Dichorhavirus	NEG	POS	NEG	NT	NEG	Inconsistent				IC to TN			1		False Positive, reacted with Novel Nucleo-rhabdovirus sequence	Incorrect
		OFV_Orc1	POS	POS	NEG	UD	NEG			1		TN to TN (No change)			1		False positive	Incorrect
13	CO005ac (*Dendrobium* sp.)	Dichorhavirus	NEG	NEG	NEG	NT	POS	Inconsistent				IC to TP				1	False Negative, reacted with Novel Dichorhavirus species sequence	Incorrect
		OFV-Gen	NEG	NEG	NEG	30.52	NEG				1	TP to TN		1			Negative	Correct
		OFV_Cit2	NEG	NEG	NEG	34.74	NEG				1	FP to TN		1			Negative	Correct
		OFV_Cit1	NEG	NEG	NEG	33.63	NEG				1	TP to TN		1			Negative	Correct

## Data Availability

The SRA data used in the study were deposited in the NCBI repository at https://www.ncbi.nlm.nih.gov/, under BioProject ID accession number PRJNA1158807, Temporary Submission ID: SUB14716102, scheduled to be released immediately after publication.

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
