# Peer review of "Detection and In Vivo Validation of Dichorhavirus e-Probes in Meta-Transcriptomic Data via Microbe Finder (MiFi®) Discovers a Novel Host and a Possible New Strain of Orchid Fleck Virus"

_viruses, 2025, doi:10.3390/v17030441_

Round 1
Reviewer 1 Report
Comments and Suggestions for Authors
Ln 22. Shorten to “Dichorhavirus is one of the a recently..”
Ln 56. Add comma after “Among these…”
Ln 74. Suggest replacing “strict” with complex
Ln 326. Clarify “length size”
Figure 2. Difficult to differentiate e-probe specificity and gene annotation
Ln 445-449. Add italics for genus and species
Table 6. Move to Results section.
Author Response
Comments and Suggestions for Authors By Reviewer 1:
Ln 22. Shorten to “Dichorhavirus is one of the a recently..”
Done.
Ln 56. Add comma after “Among these…”
Done.
Ln 74. Suggest replacing “strict” with complex
Modified.
Ln 326. Clarify “length size”
Added.
Figure 2. Difficult to differentiate e-probe specificity and gene annotation
Modified the figure 2 and provided detailed caption.
Ln 445-449. Add italics for genus and species
Added.
Table 6. Move to Results section.
In our opinion, the correct cited place for Table 6 is the Discussion section and that is why we did not move the Table 6 citation in the result section

Reviewer 2 Report
Comments and Suggestions for Authors
The Paper by Avijit Roy et al. titled "Detection and In-vivo Validation of Dichorhavirus e-Probes in Meta-Transcriptomic Data via Microbe Finder (MiFi®) Discovers a Novel Host and a Possible New Strain of Orchid Fleck Virus" presents an in-depth study on the detection and validation of Dichorhavirus e-probes using high-throughput sequencing (HTS) and the Microbe Finder (MiFi®) platform. The study successfully identified a novel host and a potential new strain of Orchid Fleck Virus (OFV). The research is well-structured, and presents significant advancements in viral detection techniques.
One of the key strengths of this paper is its innovative approach. The use of EDNA technology integrated with the MiFi® platform enhances the efficiency and accuracy of virus detection. The research question is clearly stated, and the study follows a logical progression from hypothesis to validation. Additionally, the methodology is robust, employing rigorous bioinformatics and laboratory validation techniques such as RT-qPCR and HTS, ensuring reliable results. The study effectively demonstrates the presence of known hosts and identifies a new host and a possible new strain, underscoring the effectiveness of the methodology. Furthermore, the findings have practical implications for plant disease diagnostics and biosecurity.
While the results are well-documented, the discussion section could go more deeper into the broader implications of the findings, particularly their impact on agriculture and ecology. Additionally, a direct comparison with other virus detection methods in terms of sensitivity, specificity, and cost-effectiveness would provide more context and strengthen the study.
The materials and methods section is well-detailed, allowing for replication of the study. The choice of MiFi® and e-probe curation is well-justified, with thorough explanations of the processes involved. Phylogenetic analysis and RT-qPCR validation are appropriately executed, although acknowledging potential limitations of these methods would be beneficial.
The results are well-organized and provide strong evidence supporting the study’s conclusions.
The writing is generally clear, with appropriate use of scientific language.
The references cited in the paper are relevant and recent, supporting the research effectively.
In conclusion, this is a high-quality study that significantly contributes to plant virus diagnostics. Minor revisions to improve clarity. With these refinements, the paper can have an even greater impact and readability and is highly recommended for publication
Author Response
Comments and Suggestions for Authors by Reviewer 2
While the results are well-documented, the discussion section could go more deeper into the broader implications of the findings, particularly their impact on agriculture and ecology.
although acknowledging potential limitations of these methods would be beneficial
Above two points are addressed by adding a paragraph on advantage and limitation of e-probe diagnostic at the end of discussion section (Lns 677-702) and two additional sentences (Lns 718-721) in the conclusion.
Additionally, a direct comparison with other virus detection methods in terms of sensitivity, specificity, and cost-effectiveness would provide more context and strengthen the study.
As the current article is focused on validation of e-probe diagnostic by comparing gold standard RT-qPCR assay and confirming the results by HTS. Comparison between sensitivity and specificity amongst these methods were showed in the Table 5 and 6 but did not show the comparison between the cost-effective ness with RT-qPCR assay. As for diagnosis to utilize the curated e-probe, we need the HTS data. We have mentioned in the manuscript (version 1) that e-probe diagnosis is a time saving and cost-effective method than the complete metagenomic analysis where we need an expert in computation biology. Therefore, we did not add any sentences in the revised manuscript.
